# CEP signaling coordinates plant immunity with nitrogen status

Jakub Rzemieniewski [1], Henriette Leicher[1], Hyun Kyung Lee[2], Caroline Broyart[2], Shahran Nayem[3,4], Christian Wiese [1,7], Julian Maroschek[1], Zeynep Camgöz[1], Vilde Olsson Lalun[5], Michael Anthony Djordjevic [6], A. Corina Vlot[3,4], Ralph Hückelhoven [1], Julia Santiago [2] & Martin Stegmann [1,8] ✉

Plant endogenous signaling peptides shape growth, development and adaptations to biotic and abiotic stress. Here, we identify C-TERMINALLY ENCODED PEPTIDEs (CEPs) as immune-modulatory phytocytokines in *Arabidopsis thaliana*. Our data reveals that CEPs induce immune outputs and are required to mount resistance against the leaf-infecting bacterial pathogen *Pseudomonas syringae* pv. *tomato*. We show that effective immunity requires CEP perception by tissue-specific CEP RECEPTOR 1 (CEPR1) and CEPR2. Moreover, we identify the related RECEPTOR-LIKE KINASE 7 (RLK7) as a CEP4-specific CEP receptor contributing to CEP-mediated immunity, suggesting a complex interplay of multiple CEP ligands and receptors in different tissues during biotic stress. CEPs have a known role in the regulation of root growth and systemic nitrogen (N)-demand signaling. We provide evidence that CEPs and their receptors promote immunity in an N status-dependent manner, suggesting a previously unknown molecular crosstalk between plant nutrition and cell surface immunity. We propose that CEPs and their receptors are central regulators for the adaptation of biotic stress responses to plant-available resources.

Receptor kinases (RKs) sense external and internal cues to control multiple aspects of plant physiology, ranging from growth and development to plant immunity and abiotic stress tolerance. RKs can serve as pattern recognition receptors (PRRs) to detect microbe-associated molecular patterns (MAMPs) and activate pattern-triggered immunity (PTI). An example is the *Arabidopsis thaliana* (hereafter Arabidopsis) leucine-rich repeat RK (LRR-RK) FLAGELLIN SENSITIVE 2 (FLS2), which forms a receptor complex with BRASSINOSTEROID INSENSITIVE 1-ASSOCIATED RK 1 (BAK1) upon perception of a 22 amino acid epitope derived from bacterial flagellin (flg22) to activate PTI[1-4]. Plants also perceive endogenous peptides to regulate various

aspects of their physiology[5]. Notably, specific peptides, known as phytocytokines, play an important role in controlling plant immunity, with their expression or secretion often being modulated upon PTI activation[6]. Beyond their role in defense, phytocytokines also frequently influence other physiological processes, including growth and development[7,8]. Examples are GOLVEN2 (GLV2) peptides which are perceived by ROOT MERISTEM GROWTH FACTOR INSENSITIVE 3 (RGI3) to modulate PRR stability and RAPID ALKALINIZATION FACTORs (RALFs) that are sensed by the malectin RK (MLRK) FERONIA (FER) to control PRR nanoscale dynamics at the plasma membrane and MAMP-induced PRR-BAK1 complexes for PTI initiation[9-12]. GLV2 also

[1]Phytopathology, TUM School of Life Sciences, Technical University of Munich, Freising, Germany. [2]The Plant Signaling Mechanisms Laboratory, Department of Plant Molecular Biology, University of Lausanne, Lausanne, Switzerland. [3]Helmholtz Zentrum Munich, Institute of Biochemical Plant Pathology, Neuherberg, Germany. [4]Chair of Crop Plant Genetics, Faculty of Life Sciences: Food, Nutrition and Health, University of Bayreuth, Kulmbach, Germany. [5]Department of Biosciences Section for Genetics and Evolutionary Biology, Department of Biosciences, University of Oslo, Oslo, Norway. [6]Division of Plant Science, Australian National University, Acton, ACT, Australia. [7]Present address: Biotechnology of Natural Products, TUM School of Life Sciences, Technical University of Munich, Freising, Germany. [8]Present address: Institute of Botany, Molecular Botany, Ulm University, Ulm, Germany. ✉e-mail: martin.stegmann@tum.de

controls hypocotyl gravicurvature[13] and RALF perception by FER and other MLRKs affects several aspects of plant growth, development and reproduction, suggesting that endogenous peptides coordinate these processes with stress responses[14–19].

Immune-modulatory peptides are often transcriptionally upregulated in response to MAMP perception, including SERINE-RICH ENDOGENOUS PEPTIDES (SCOOPs) and SMALL PHYTOCYTOKINES REGULATING DEFENSE AND WATER LOSS (SCREWs)/CTNIPs[20–22]. Yet, *GLV2* transcription is not induced by biotic stress[9] and promotes immunity, suggesting that immunity-dependent transcriptional regulation is not a prerequisite for phytocytokine function. Phytocytokines and other endogenous peptides further regulate a multitude of abiotic stress responses, including adaptation to high salinity, drought and nutrient deprivation, indicating that they can integrate multiple external and internal cues to safeguard plant health[23–25]. Yet, how different peptide-mediated pathways are coordinated remains largely unknown.

Here, we identified C-TERMINALLY ENCODED PEPTIDES (CEPs) as phytocytokines in Arabidopsis. CEPs are important for sucrose-dependent lateral root growth, root system architecture, systemic nitrogen (N)-demand signaling and promotion of root nodulation, but a function in plant immunity remained unknown[24,26–32]. We show that the unusual class I CEP peptide CEP4 induces immune responses. We found that *CEP4* and other *CEPs* are expressed in shoots and perceived by canonical CEP receptors CEPR1 and CEPR2 to mount effective cell surface immunity. *CEPR1* and *CEPR2* show tissue-specific expression patterns, suggesting CEP sensing in distinct tissues spatially cooperates to control plant immunity. Yet, CEP4-induced responses also require the CEPR-related RECEPTOR-LIKE KINASE 7 (RLK7), which we identified as a CEP4-specific CEP receptor with widespread expression in leaves. Importantly, we now show that a reduction in seedling N availability promotes flg22-induced MAPK activation and bacterial resistance in a CEPR-CEP-dependent manner, suggesting that CEPs coordinate a previously unknown cross-talk between cell surface immunity and plant nutrition.

## Results

### CEPs are phytocytokines

We sought to identify phytocytokines regulating growth and immunity in Arabidopsis and screened publicly available transcription data of known growth-regulatory plant peptide families for members with differential expression after elicitor treatment. With this approach, we recently identified GLV2 as a phytocytokine modulating PRR stability through RGIs[9]. We noticed that a specific member of the CEP family, *CEP4*, showed differential expression upon flg22 treatment with a moderate downregulation in an *asr3* mutant background, a transcriptional repressor of flg22-induced genes (Supplementary Fig. 1A)[33]. Using RT-qPCR, we also observed a mild flg22-induced *CEP4* downregulation in Col-0 seedlings compared to the mock control (Supplementary Fig. 1B). At 4 h of mock treatment, *CEP4* expression was lower compared to 0-hour mock samples, suggesting some degree of circadian rhythm-related regulation (Supplementary Fig. 1B). Most *CEPs* are primarily expressed in root tissue[34]. As expected, we detected much higher levels of *CEP4* transcript in the root compared to the shoot (Supplementary Fig. 1C). We also examined whether *CEP4* expression is differentially regulated after flg22 treatment depending on the tissue type. Interestingly, 1 h after flg22 treatment, *CEP4* levels in the shoot slightly increased, while *CEP4* transcripts in the root were mildly downregulated, raising the question of tissue-specific regulation of *CEP4* expression (Supplementary Fig. 1C). Arabidopsis encodes 12 class I CEPs, which can be distinguished by sequence differences in their peptide domain[34,35]. CEPs are produced from larger peptide precursors that carry one to five predicted mature CEP domains in their sequence and an N-terminal signal peptide for secretion[34]. While CEP4 is classified as a class I CEP, it has an unusual structure compared

to other family members, carrying only two proline residues in its peptide domain, unlike several characterized typical class I CEPs such as CEP1 and CEP3 (Supplementary Fig. 1D)[34]. We observed that most class I *CEPs* also exhibited higher transcript levels in the roots, with mild tissue-specific differences in expression pattern following flg22 treatment (Supplementary Fig. 1C).

To test whether CEP4 may be involved in immunity, we generated constitutive overexpression lines using a full-length *CEP4* precursor sequence (*35S::CEP4*, Supplementary Fig. 2A) and noticed that these lines showed increased resistance to infection by *Pseudomonas syringae* pv. *tomato (Pto)* lacking the effector molecule coronatine (*Pto*^COR-), which is routinely used to assess PTI-associated disease resistance phenotypes (Fig. 1A)[36,37]. The same lines were also more resistant to infection with the fully virulent, wild-type *Pto* DC3000 strain (Supplementary Fig. 2B). As a positive control for *Pto* infection we used the hypersusceptible *bak1-5/bkk1-1* double mutant[38].

The majority of mature CEPs previously identified are 15mer peptides with an N-terminal aspartate (D), a C-terminal Histidine (H) and hydroxylated prolines[24,34]. We synthesized a 16mer peptide with both proline residues hydroxylated and the N- and C-terminal D and H residue, respectively, DAFRHypTHQGHypSQGIGH, to test whether it triggered or modulated immune responses. CEP4 application activated dose-dependent PTI outputs, including the cellular influx of calcium ions in a Col-0 line expressing the calcium reporter Aequorin (Col-0^AEQ)[39], the activation of MITOGEN-ACTIVATED PROTEIN KINASEs (MAPKs), ethylene production and expression of the PTI marker gene *FLAGELLIN-INDUCED RECEPTOR KINASE 1* (*FRK1*) in Col-0 seedlings (Fig. 1B–E, Supplementary Fig. 3A). CEP4-induced calcium influx was detectable in the low nanomolar range of CEP4 concentration (Supplementary Fig. 3A) and MAPK phosphorylation was detected at concentrations of 100 nM, yet the magnitude of response was weaker compared to flg22 (Supplementary Fig. 3B). *FRK1* transcript accumulation and calcium influx activated by flg22 treatment was much stronger compared to CEP4 in whole seedlings (Supplementary Fig. 3C, D). However, CEP4 induced *FRK1* expression in a similar range as previously described elicitors, suggesting biological relevance[40]. Moreover, CEP4 triggered nuclear YFP fluorescence in the vasculature of *pFRK1::NLS-3xmVenus* seedlings, suggesting some degree of tissue specificity of CEP4-induced immune outputs[41] (Fig. 1F, Supplementary Fig. 3F). Finally, CEP4 treatment resulted in seedling growth inhibition (SGI) and systemic resistance to *Pto* DC3000 infection (Fig. 1G, H). We also tested a 15mer peptide lacking the N-terminal D residue (AFRHypTHQGHypSQGIGH), which triggered a dose-dependent calcium influx with no significant differences to the response elicited by the 16mer peptide (Supplementary Fig. 3A). We next tested whether other class I CEPs can activate PTI responses. Indeed, CEP1 and one peptide derived from CEP9 (CEP9.5), which carries five CEP domains in its precursor sequence[35], were able to trigger calcium influx in seedlings but only at higher concentrations of 10 µM (Supplementary Fig. 3E). However, CEP1 induced nuclear YFP fluorescence in the vasculature of *pFRK1::NLS-3xmVenus* lines at similar concentrations as CEP4 (Supplementary Fig. 3F), suggesting that likely several CEPs can trigger immune responses with CEP4 being a very potent family member.

To confirm the role of CEPs in plant immunity and because of anticipated genetic redundancy, we generated a *cep6x* mutant by CRISPR-Cas9 in which *CEP4*, as well as the five additional class I CEPs *CEP1-CEP3*, *CEP6* and *CEP9* were mutated to predictable loss of function (CRISPR alleles *cep1.1*, *cep2.1*, *cep3.1*, *cep4.1*, *cep6.1* and *cep9.1*, Supplementary Fig. 2C). The resulting *cep6x* mutant had no obvious morphological defects (Supplementary Fig. 2D) but showed compromised resistance to spray infection with *Pto*^COR- and *Pto* DC3000, confirming that CEPs are important for antibacterial resistance (Fig. 1I, Supplementary Fig. 4A). To overcome the impact of different tissue-specific CEPs[24,34] that might cooperate to mount disease resistance, we partially complemented the *Pto* hypersusceptibility phenotype of

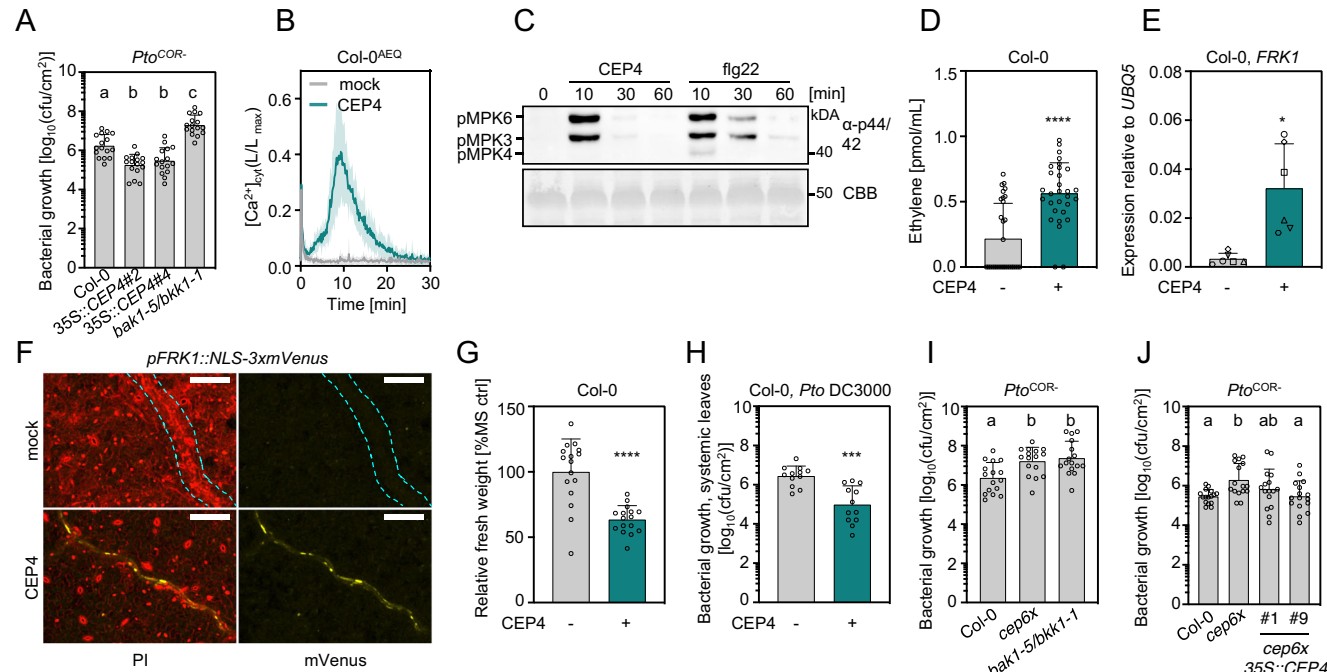

**Fig. 1 | CEPs induce immune responses and are important determinants of plant immunity. A** Colony forming units (cfu) of $Pto^{COR-}$ 3 days post inoculation (3 dpi) upon spray infection; n = 16 pooled from four independent experiments with mean ± SD (one-way ANOVA, Tukey post-hoc test; a-b, $p < 0.005$; a/b-c $p < 0.0001$). **B** Kinetics of cytosolic calcium concentrations ($[Ca^{2+}]_{cyt}$) in Col-0$^{AEQ}$ seedlings upon mock (ddH$_2$O) or CEP4 (1 µM) treatment; n = 8, mean ± SD. **C** MAPK activation in Col-0 upon CEP4 (1 µM) or flg22 (100 nM) treatment for the indicated time. Western blots were probed with α-p44/42. Size marker is indicated. CBB = Coomassie brilliant blue. **D** Ethylene concentration in Col-0 leaf discs 3.5 h upon mock (ddH$_2$O) or CEP4 (1 µM) treatment; n = 30 pooled from six independent experiments with mean ± SD (Mann-Whitney U test, $^{****} p < 0.0001$). **E** RT-qPCR of $FRK1$ in seedlings upon CEP4 (1 µM) or mock (ddH$_2$O) treatment for 4 h. Housekeeping gene $UBQ5$; n = 6, mean ± SD with different symbols showing independent experiments

(Welch's $t$-test, $^{*}p = 0.0113$). **F** NLS-3xmVenus signal in p$FRK1$::NLS-3xmVenus upon mock (ddH$_2$O) or CEP4 (100 nM) treatment for 16 h. Cyan dotted line indicates vasculature. PI = propidium iodide, scale bar = 100 µm. **G** Relative fresh weight (as percent of ½ MS medium control = % MS ctrl) of five-day-old seedlings treated with CEP4 (1 µM) for seven days; n = 16 (Mann-Whitney U test, $^{****} p < 0.0001$). **H** cfu of $Pto$ DC3000 (4 dpi) in distal leaves upon local CEP4 (1 µM) or mock (ddH$_2$O) pre-treatment; n = 12 pooled from three independent experiments with mean ± SD (Welch's $t$-test, $^{***} p = 0.0002$). **I** cfu of $Pto^{COR-}$ (3 dpi) upon spray infection; n = 16 pooled from four independent experiments with mean ± SD (one-way ANOVA, Tukey post-hoc test, a-b $p \leq 0.01$). **J**) cfu of $Pto^{COR-}$ (3dpi) upon spray infection; n = 16 pooled from four independent experiments with mean ± SD (one-way ANOVA, Tukey post-hoc test, a-b $p < 0.05$). All experiments were performed at least three times in independent biological repeats with similar results.

$cep6x$ using a full-length $CEP4$ driven by the constitutive 35S promoter (Fig. 1J, Supplementary Fig. 2E, 4A). We also tested a $cep5x$ mutant (CRISPR alleles $cep1.2$, $cep2.2$, $cep3.2$, $cep6.2$ and $cep9.2$) in which CEP4 is wild-type (Supplementary Fig. 2C, D). This mutant was less suscep-tible to spray infection with $Pto^{COR-}$ and the wild-type $Pto$ DC3000 compared to $cep6x$ (Supplementary Fig. 4B, C). This indicates an important contribution of CEP4 with an additive effect of CEP1-3, CEP6 and CEP9 to mount robust resistance.

## CEPR1 and CEPR2 are CEP4 receptors and central regulators of plant immunity

CEPs bind to CEPR1 and CEPR2, two LRR-RLKs from LRR subfamily XI[24,42]. Genetically, CEPR1 is predominantly required for class I CEP perception during root growth-related responses[26–28,30,43,44]. We tested whether $CEPR1$ and $CEPR2$ are also involved in bacterial immunity. We did not observe a significant change in resistance to spray-inoculated $Pto^{COR-}$ in $cepr1-3$ and $cepr2-4$ single mutants (Fig. 2A, Supplementary Fig. 5A). We generated a $cepr1-3$ $cepr2-4$ ($cepr1-3/2-4$) mutant by genetic crossing and this double mutant was more susceptible to spray infection with $Pto^{COR-}$ (Fig. 2A). We also generated a new $cepr1$ $cepr2$ double mutant by CRISPR-Cas9 in a Col-0$^{AEQ}$ background (CRISPR alleles $cepr1.1^{AEQ}$, $cepr2.1^{AEQ}$, hereafter $cepr1/2^{AEQ}$) (Supplementary Fig. 5B). Similar to $cepr1-3/2-4$, $cepr1/2^{AEQ}$ was more susceptible to spray-inoculated $Pto^{COR-}$ (Fig. 2A), further confirming a role of $CEPR1$ and $CEPR2$ in antibacterial resistance. These data suggest that $CEPR1$ and $CEPR2$ may control immunity redundantly.

We then tested whether CEP4 perception depends on CEPR1 and/or CEPR2. Using SGI as a readout, we noticed that $cepr1-3$ was insensitive and two $cepr2$ alleles ($cepr2-3$ and $cepr2-4$) were insen-sitive and less sensitive to CEP4 treatment, respectively (Supplemen-tary Fig. 5C). The $cepr1-3/2-4$ mutant and the previously reported No-0 $cepr1-1$ $cepr2-1$ double mutant also did not respond to CEP4 in SGI experiments (Fig. 2B, Supplementary Fig. 5D). Similarly, the $cepr1-3/2-4$ double mutant was largely insensitive to CEP4-induced systemic resistance (Fig. 2C). Collectively, these data suggest that both CEPR1 and CEPR2 are involved in CEP4 perception. We then tested whether CEP4 can directly bind to the ectodomain (ECD) of CEPR1 and/or CEPR2. We expressed CEPR1$^{ECD}$ and CEPR2$^{ECD}$ in $Trichoplusia$ $ni$ Tnao38 cells and purified them for quantitative binding experiments. We analyzed protein quality by Coomassie stain and size exclusion chromatography (SEC, Supplementary Fig. 6A, B). Unfortunately, CEPR1$^{ECD}$ aggregated, as indicated by the early elution of the bulk sample during SEC analysis (~10 min, Supplementary Fig. 6A). Nevertheless, we obtained good quality protein for CEPR2$^{ECD}$ with a single SEC elution peak at ~13 min, which we subsequently tested for quantitative binding to CEP4 using isothermal titration calorimetry (ITC) (Supplementary Fig. 6A). CEP4, but not a scrambled control (CEP4$^{scr}$), directly bound to CEPR2$^{ECD}$ with a $K_D$ of 15.7 µM (± 4.5 µM) (Fig. 2D, E, G, Supplementary Fig. 6C). CEP1, which was previously shown to bind to CEPR1 and CEPR2[24] also bound to CEPR2$^{ECD}$ with a $K_D$ of 9.3 µM (± 0.6 µM) (Fig. 2F, G, Supplementary Fig. 6C). These data are in range with previously reported peptide-LRR-RK binding

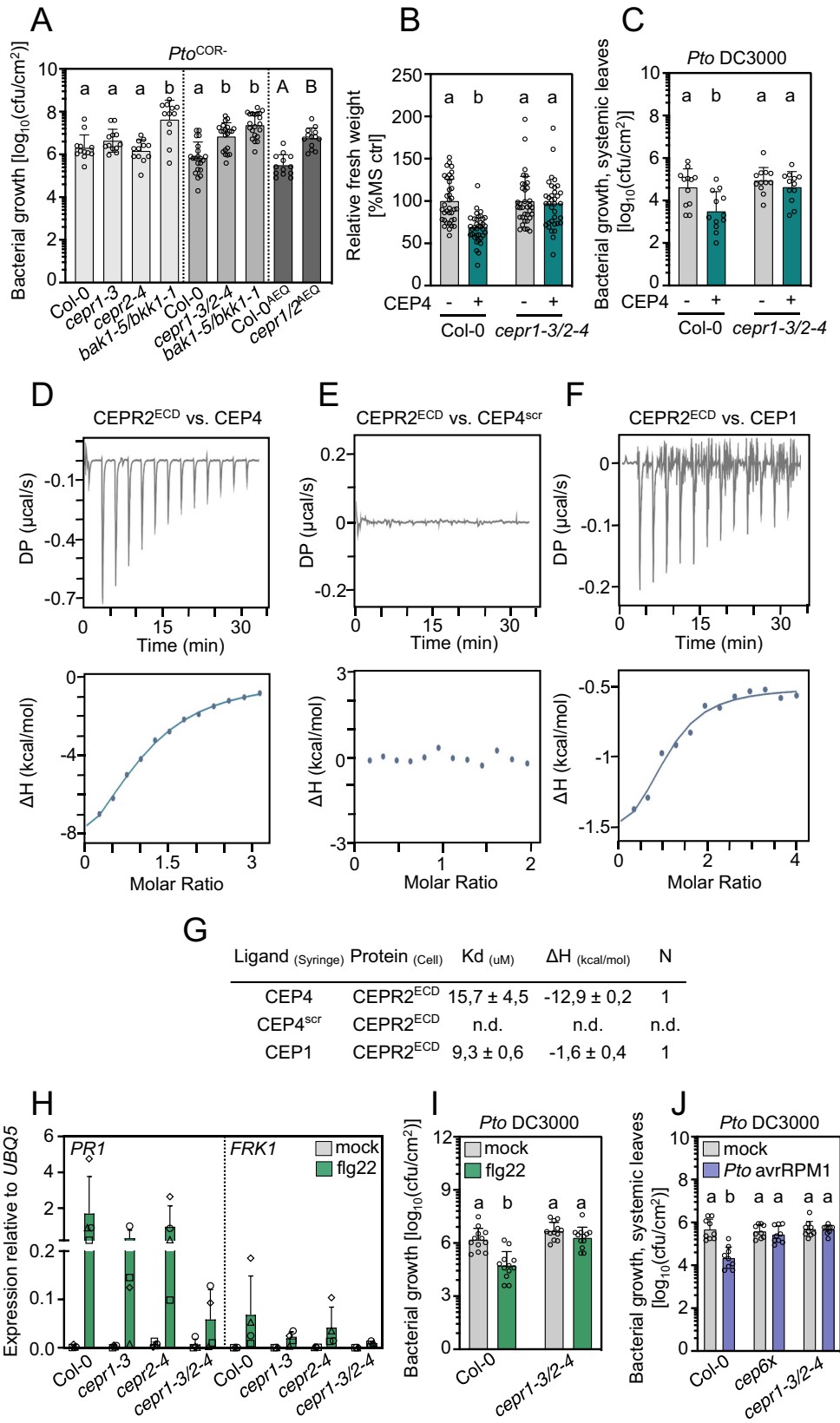

affinities obtained by ITC and suggest that CEPR2 is a bona fide CEP4 receptor[45–47].

We were then interested in characterizing a possible role of CEPR1 and CEPR2 in FLS2-mediated signaling. We did not observe strong differences in flg22-induced ethylene accumulation *in cepr1-3/2-4* (Supplementary Fig. 7A). Of note, flg22-induced ethylene production

was higher in *cepr1-3/2-4*, but basal ethylene production was enhanced in this mutant background, making the result difficult to interpret. This is interesting in light of a previous report showing that the *Medicago truncatula* CEP1-CRA2 (CEPR1 orthologue) pathway negatively regulates ethylene signaling during root nodule symbiosis[48]. Interestingly though, flg22-induced *FRK1* and *PR1* expression were reduced in adult

**Fig. 2 | CEPR1 and CEPR2 are CEP4 receptors and are important determinants of plant immunity. A** cfu of $Pto^{COR-}$ (3 dpi) upon spray infection. The dotted line indicates different experiments; $n = 12$, 21 and 13 pooled from three, five and three experiments, respectively, with mean ± SD. Statistical analysis was performed separately for each group (one-way ANOVA, Dunnett's post-hoc test, a-b $p < 0.0001$, two-tailed Student's $t$-test, A-B $p < 0.0001$). **B** Relative fresh weight of five-days-old seedlings treated with CEP4 (1 μM) for seven days; Col-0 ($n = 36$), *cepr1-3/2-4* ($n = 35$) pooled from three experiments with mean ± SD (Kruskal-Wallis, Dunn's post-hoc test, a-b $p < 0.0001$). **C** cfu of *Pto* DC3000 (4 dpi) in distal leaves following local CEP4 (1 μM) or mock (ddH₂O) pre-treatment; $n = 12$ pooled from three experiments with mean ± SD (one-way ANOVA, Tukey post-hoc test, a-b $p < 0.01$). **D** CEP4, **E** CEP4$^{scr}$ and **F** CEP1 were titrated into a solution containing CEPR2$^{ECD}$ in ITC cells. Top: raw data thermogram; bottom: fitted integrated ITC data curves. DP = differential power between reference and sample cell; ΔH = enthalpy change. **G** ITC table summarizing CEPR2$^{ECD}$ vs CEP4/CEP4$^{scr}$/CEP1 as means ± SD of two experiments. The dissociation constant ($K_d$) indicates receptor-ligand binding affinity. N indicates reaction stoichiometry ($n = 1$ for 1:1 interaction). **H** RT-qPCR of *PR1* and *FRK1* in adult leaves after treatment with flg22 (1 μM) or mock (ddH₂O) for 24 h. Housekeeping gene *UBQ5*; $n = 4$, mean ± SD with different symbols showing independent experiments. **I** cfu of *Pto* DC3000 (3 dpi) in leaves following flg22 (1 μM) or ddH₂O pre-treatment; $n = 12$ pooled from three experiments with mean ± SD (one-way ANOVA, Tukey post-hoc test, a-b $p < 0.0001$). **J** cfu of *Pto* DC3000 (4 dpi) in distal leaves following local infection with *Pto avrRPM1* or 10 mM MgCl₂; $n = 9$ pooled from three independent experiments with mean ± SD (one-way ANOVA, Tukey post-hoc test, a-b $p < 0.0001$). Experiments in **A**–**C** and **H**–**J** were performed at least three times in independent biological repeats. Experiments in **D**–**F** were repeated in two independent technical repeats with similar results.

*cepr1-3* or *cepr2-4* plants, which was pronounced in *cepr1-3/2-4* (Fig. 2H). Moreover, *cepr1-3/2-4* showed compromised flg22-induced resistance to *Pto* DC3000 infection (Fig. 2I). The flg22-induced seedling growth inhibition was unaffected in *cepr1-3/2-4* (Supplementary Fig. 7B), suggesting that CEPR1 and CEPR2 are selectively required for specific flg22-induced outputs associated with antibacterial defense. Since local CEP4 pre-treatment can increase resistance to bacterial infection in distal tissues (Fig. 1H, Fig. 2C), we were interested in testing whether CEP-CEPR1/2 signaling may be important for systemic acquired resistance (SAR). To induce SAR, we used a *Pto* strain producing the effector avrRPM1 (*Pto* avrRPM1), which is recognized by Col-0 RESISTANCE TO PSEUDOMONAS SYRINGAE 1, to activate effector-triggered immunity and consequently SAR[49,50]. Local inoculation of *Pto* avrRPM1 and subsequent infection of systemic tissue with virulent *Pto* revealed that *cepr1-3/2-4*, as well as *cep6x*, were strongly compromised in *Pto* avrRPM1-triggered SAR (Fig. 2J). Interestingly, mock-treated *cep6x* (Fig. 2J) and *cepr1-3/2-4* mutants (Fig. 2C, I, J) do not show enhanced *Pto* DC3000 growth compared to Col-0, suggesting that CEP-CEPR signaling primarily regulates immunity to tissue invasion, an effect that is bypassed by syringe infiltration in this experimental setup. Collectively, these data show that CEP-CEPR branch plays a central role in regulating cell surface immunity and SAR in Arabidopsis.

## RLK7 is a CEP4-specific CEP receptor

To further characterize the role of CEPR1 and CEPR2 in CEP4-induced signaling, we tested early CEP4-triggered responses in *cepr1-3/2-4* and *cepr1/2*$^{AEQ}$. To our surprise, we found that *cepr1/2*$^{AEQ}$ did not show compromised calcium influx upon CEP4 treatment (Fig. 3A). Similarly, CEP4-induced MAPK activation was not impaired and CEP4-induced *FRK1* expression was only partially reduced in the double *cepr1-3/2-4* knock-out (Fig. 3B, Supplementary Fig. 8A). These results suggested that other receptor(s) may be involved in CEP4 perception. CEPR1 and CEPR2 are phylogenetically close to IKU2, which is involved in seed size regulation, and RLK7, which plays a role in controlling germination speed, lateral root formation and salt stress adaptation[25,42,51–53]. RLK7 also senses endogenous PAMP-INDUCED PEPTIDES (PIPs) to regulate PTI and resistance to the fungal wilt pathogen *Fusarium oxysporum* and *Pto*[54,55]. We tested whether *iku2-4*, *rlk7-1* and *rlk7-3* are compromised in CEP4 perception. The *iku2-4* mutant showed unaltered CEP4-induced ethylene accumulation, but this response was impaired in *rlk7-1* and *rlk7-3* (Supplementary Fig. 8B). Similarly, *rlk7-1* and *rlk7-3* showed strongly reduced CEP4-induced MAPK activation (Fig. 3C). We also generated an *rlk7/iku2*$^{AEQ}$ mutant by CRISPR-Cas9 in a Col-0$^{AEQ}$ background (CRISPR alleles *rlk7.1*$^{AEQ}$ and *iku2.1*$^{AEQ}$, Supplementary Fig. 8C). The *rlk7/iku2*$^{AEQ}$ line showed compromised CEP4-induced calcium influx (Supplementary Fig. 8D). Two additional *rlk7*$^{AEQ}$ single mutants generated by CRISPR-Cas9 (CRISPR alleles *rlk7.2*$^{AEQ}$, *rlk7.3*$^{AEQ}$ Supplementary Fig. 8C) were also compromised in CEP4-triggered calcium influx (Fig. 3D). Yet, residual CEP4 activity remained in *rlk7* mutants,

both for CEP4-induced MAPK activation and calcium influx (Fig. 3C, D). To resolve this, we generated a CRISPR *cepr1 cepr2 rlk7*$^{AEQ}$ (hereafter *cepr1/2/rlk7*$^{AEQ}$ or *c1/c2/r7*$^{AEQ}$) triple mutant using the *cepr1/2*$^{AEQ}$ background (CRISPR allele *rlk7.4*$^{AEQ}$ in *cepr1/2*$^{AEQ}$, Supplementary Fig. 5B, 8C), which showed abolishment of CEP4-induced calcium influx and MAPK activation (Fig. 3E, F). This suggests that RLK7, CEPR1 and CEPR2 each participate in mounting a full CEP4 response, with RLK7 playing a predominant genetic role. The *cepr1/2/rlk7*$^{AEQ}$ mutant also showed abolished CEP4-induced resistance to *Pto* DC3000 infection (Fig. 3G). Both *rlk7*$^{AEQ}$ and *cepr1/2/rlk7*$^{AEQ}$ were also insensitive to PIP1 in MAPK activation (Supplementary Fig. 8E), consistent with RLK7's function as a PIP receptor[55]. Interestingly, *rlk7-1* and *rlk7-3* were not significantly affected in CEP4-induced SGI (Supplementary Fig. 8F), suggesting that certain CEP4 responses require selective specificity for one of the three CEP receptors. Moreover, CEP1 and CEP9.5-induced calcium influx was abolished in *cepr1/2*$^{AEQ}$ and unaltered in *rlk7/iku2*$^{AEQ}$ or *rlk7.2*$^{AEQ}$, respectively, indicating that CEPR1/2 are required for the early responses triggered by these canonical class I CEPs, which again show differential receptor requirements (Fig. 3H, Supplementary Fig. 8G).

We next used ITC to test whether CEP4 can bind to the ectodomain of RLK7 (RLK7$^{ECD}$). Similar to CEPR2$^{ECD}$, we obtained good quality proteins for RLK7$^{ECD}$ eluting with a main single peak at -13 min during SEC analysis (Supplementary Fig. 6A, B). CEP4 bound to RLK7$^{ECD}$ with a $K_D$ of 9 μM (± 4.9 μM) (Fig. 3I, L, Supplementary Fig. 6C), similar to CEPR2$^{ECD}$ (Fig. 2D, G). We also tested binding of PIP1 to RLK7$^{ECD}$, a described RLK7 ligand[55]. RLK7$^{ECD}$ bound PIP1 with a higher affinity ($K_D$ 500 nM ± 60 nM) (Fig. 3J, L, Supplementary Fig. 6C). Importantly, consistent with unaltered CEP1-induced responses in *rlk7/iku2*$^{AEQ}$ (Fig. 3H), CEP1 did not bind to RLK7 (Fig. 3K, Supplementary Fig. 6C). These data suggest that in addition to PIP perception, RLK7 can also function as a CEP4-specific CEP receptor.

The *cepr1/2*$^{AEQ}$ mutants show compromised resistance to *Pto*$^{COR-}$ (Fig. 2A), similar to previously published *rlk7* single mutants[54]. Further mutation of *rlk7* in *cepr1/2*$^{AEQ}$ background did not significantly enhance this phenotype (Supplementary Fig. 8H). When using the fully virulent *Pto* DC3000 strain, *cepr1/2*$^{AEQ}$ was moderately more susceptible (Fig. 3M), whereas *rlk7* was not[54]. Interestingly, the *cepr1/2/rlk7*$^{AEQ}$ triple mutant showed significantly increased susceptibility to *Pto* DC3000 compared to *cepr1/2*$^{AEQ}$ (Fig. 3M), indicating that all three CEP receptors mount full antibacterial resistance.

## CEP-CEP receptor signaling promotes local immunity against *Pto*

Despite the generally lower expression levels of class I *CEPs* in aboveground tissues (Supplementary Fig. 1C), the *cep6x* mutant showed increased susceptibility to bacterial infection in spray-inoculated leaves (Fig. 1I, J). Considering that CEPs act as mobile, root-to-shoot transmitters of N starvation signals[24], we hypothesized that the root-expressed *CEPs* may also contribute to leaf immunity against *Pto* systemically. To test whether root or shoot expression of *CEPs* is required

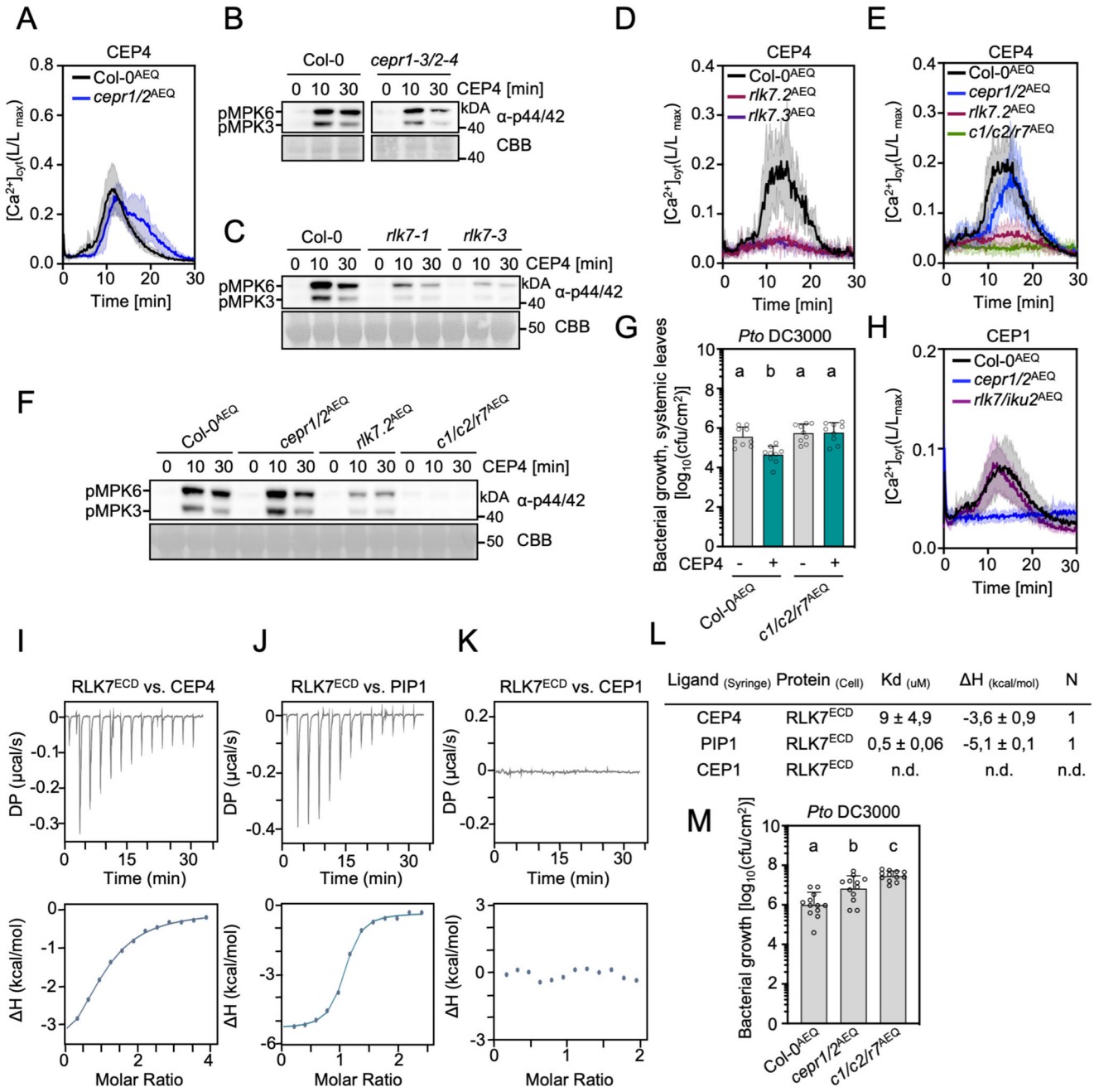

**Fig. 3 | RLK7 is an additional CEP4 receptor. A** $[Ca^{2+}]_{cyt}$ kinetics in seedlings upon CEP4 (1 µM) treatment; $n = 6$, mean ± SD. **B, C** MAPK activation upon CEP4 (1 µM) treatment for the indicated time. **D, E** $[Ca^{2+}]_{cyt}$ kinetics in seedlings upon CEP4 treatment (1 µM); $n = 3$ (D) or $n = 4$ (E), respectively, mean ± SD. $c1/c2/r7^{AEQ} = cepr1/2/rlk7^{AEQ}$. **F** MAPK activation upon CEP4 (1 µM) treatment for the indicated time. **G** cfu of *Pto* DC3000 (4 dpi) in distal leaves upon local CEP4 (5 µM) or mock (ddH₂O) pre-treatment; $n = 9$ pooled from three independent experiments with mean ± SD (Kruskal-Wallis, Dunn's post-hoc test, a-b $p < 0.05$). **H** $[Ca^{2+}]_{cyt}$ kinetics in seedlings upon CEP1 (10 µM) treatment; $n = 8$, mean ± SD. **I** CEP4, **J** PIP1 and **K** CEP1 were titrated into a solution containing CEPR2$^{ECD}$ in ITC cells. Top: raw data thermogram; bottom: fitted integrated ITC data curves. DP = differential power

between reference and sample cell; ΔH = enthalpy change. **L** ITC table summarizing RLK7$^{ECD}$ vs CEP4/PIP1/CEP1 as mean ± SD of two experiments. The dissociation constant ($K_d$) indicates receptor-ligand binding affinity. N indicates reaction stoichiometry ($n = 1$ for 1:1 interaction). **M** cfu of *Pto* DC3000 (3 dpi) upon spray infection; $n = 12$ pooled from three independent experiments with mean ± SD (one-way ANOVA, Tukey post-hoc test, a-b $p = 0.0021$; a-c $p < 0.0001$; b-c $p = 0.0209$). Western blots in **B, C** and **F** were probed with α-p44/42. Size marker is indicated. CBB = Coomassie brilliant blue. Experiments in **I–K** were repeated two times in independent technical repeats with similar results. Other experiments were performed at least three times in independent biological repeats with similar results, except **F**, which has been performed twice with identical results.

for immune regulation in the leaf, we performed reciprocal grafts between Col-0 and *cep6x* plants. Surprisingly, we found that *CEP* mutation in the shoot conferred increased *Pto*$^{COR-}$ susceptibility in *cep6x* (Fig. 4A). We used the hypovirulent *Pto*$^{COR-}$ in this experiment, as the *cep6x* phenotype is stronger with this bacterial strain (Fig. 1I, Supplementary Fig. 4A). Consistently, we detected weak *CEP1-CEP4, CEP6* and

*CEP9* expression in leaf tissue with *CEP4* being slightly upregulated upon *Pto* DC3000 syringe infiltration (Fig. 4B), similar to the mild *CEP4* upregulation upon flg22 treatment in seedling shoots (Supplementary Fig. 1C). We also found other class I *CEPs* to be stably expressed or mildly upregulated in leaves after *Pto* DC3000 infection (Supplementary Fig. 4D), suggesting a possible involvement in shoot immunity.

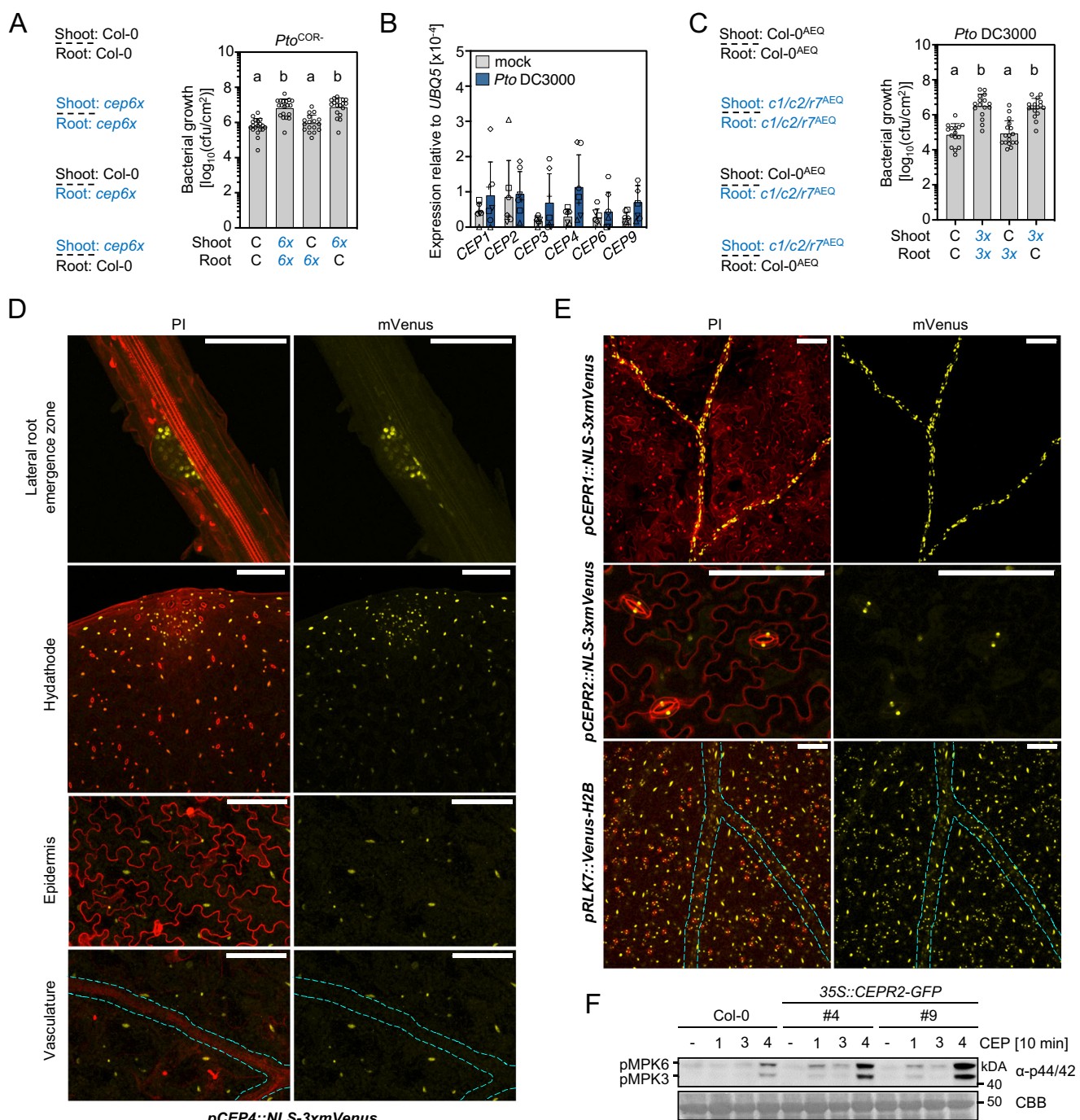

**Fig. 4 | Shoot-expressed *CEPs*, *CEPR1*, *CEPR2* and *RLK7* receptors are required for basal immunity against *Pto*. A** Cfu of *Pto*[COR-] (3 dpi) upon spray infection of reciprocally grafted Col-0 (C) and *cep6x* (*6x*) plants. S:Col-0/R:Col-0, S:*cep6x*/R:*cep6x* (*n* = 19), S:Col-0/R:*cep6x* (*n* = 17), S:*cep6x*/R:Col-0 (*n* = 19) pooled from four independent experiments with mean ± SD (Kruskal-Wallis, Dunn's post-hoc test, a-b *p* ≤ 0.001). **B** RT-qPCR analysis of *CEP* expression in mock (ddH₂O) and *Pto* DC3000-inoculated leaves 24 h post-treatment. Housekeeping gene *UBQ5*; *n* = 7, mean ± SD with different symbols showing independent experiments. **C** cfu of *Pto*[COR-] (3 dpi) upon spray infection of reciprocally grafted Col-0[AEQ] (C) and *cepr1/2/rlk7*[AEQ] (*3x*) plants. S:Col-0[AEQ]/R:Col-0[AEQ] (*n* = 15), S:*c1/2/r7*[AEQ]/R:*c1/2/r7*[AEQ] (*n* = 16), S:Col-0[AEQ]/R:*c1/2/r7*[AEQ] (*n* = 17), S:*c1/2/r7*[AEQ]/R:Col-0[AEQ] (*n* = 16) pooled from three independent experiments with mean ± SD (one-way ANOVA, Tukey post-hoc test, a-b *p* < 0.0001). **D** *pCEP4::NLS-3xmVenus* signal in the indicated plant tissues. For

imaging the lateral root emergence zone and the hydathode region, the maximum projection of Z-stacks for mVenus is merged with Z-stacked propidium iodide (PI) signal. To show the lack of mVenus signal in the vasculature, the maximum projection of Z-stacks for mVenus is merged with the same single section of PI, showing a single epidermal layer or vasculature. **E** NLS-3xmVenus or Venus-H2B signal in the leaves of indicated lines. The maximum projection of Z-stacks for mVenus is merged with Z-stacked PI signal. **F** MAPK activation upon CEP1 (1), CEP3 (3) and CEP4 (4) 1 μM treatment. Western blots were probed with α-p44/42. Size marker is indicated. CBB = Coomassie brilliant blue. The cyan-dotted line in **D** and **E** represents vasculature; scale bar = 100 μm. All experiments were performed at least three times in independent biological repeats with similar results, except **F**, which has been performed twice with identical results.

We similarly performed reciprocal grafting with Col-0[AEQ] and cepr1/2/rlk7[AEQ] mutants, which demonstrated that shoot-expressed CEP receptors are required to confer enhanced *Pto* DC3000 resistance (Fig. 4C). *Pto* DC3000 was used because the phenotype of *cepr1/2/rlk7*[AEQ] was more pronounced with this bacterial strain (Fig. 3M, Supplementary Fig. 8H). These data suggest that CEP function in the shoot is necessary for their immune-modulatory function, unlike the root-to-shoot CEP mobility required for N-demand signaling[24].

Next, we wanted to investigate spatial expression patterns of *CEP4* by generating a *pCEP4::NLS-3xmVenus* line. Consistent with previous reports, the *CEP4* promoter was not active in the main root, but in emerging lateral roots (Fig. 4D)[34]. Despite weak shoot signals for *CEP4* expression obtained by qPCR (Supplementary Fig. 1C), we found widespread *CEP4* promoter activity in seedling leaf tissue, but not in the vasculature or stomatal guard cells (Fig. 4D). This data further supports a role for leaf-expressed CEP4 in local shoot immune responses.

We next sought to determine the spatial expression patterns of *CEPR1*, *CEPR2* and *RLK7* in shoot tissue. Previous studies using promoter::β-GLUCURONIDASE lines indicated that *CEPR1* expression is restricted to the vasculature, while *CEPR2* promoter activity is more widespread[24,56]. Consistent with these findings, the *pCEPR1::NLS-3xmVenus* activity in leaves confirmed *CEPR1*'s specificity in vascular tissue, whereas *pCEPR2::NLS-3xmVenus* signals were predominantly localized to stomatal guard cells (Fig. 4E). The *pRLK7::Venus-H2B* line showed widespread promoter activity, including guard cell, vasculature, mesophyll and epidermal cells (Fig. 4E). *RLK7* showed a large overlap with *CEP4* expression in leaves, suggesting that CEP4 and RLK7 can meet in vivo to function as a receptor-ligand pair. We also observed that flg22 treatment did not change the tissue-specific promoter activity of *CEP4* or *CEPR1/CEPR2/RLK7* (Supplementary Fig. 9). Although our results point to CEPR1/CEPR2 and RLK7 being genetically required for full CEP4 sensitivity (Fig. 3E, F), only *RLK7* promoter activity showed a clear overlap with *CEP4* promoter activity in leaf tissue. Considering that the *CEP4* promoter is also active in emerging lateral roots (Fig. 4D), we examined the promoter activity of *CEPR1*, *CEPR2*, and *RLK7* in below-ground tissues. We observed a possible expressional overlap in the base of young lateral roots between *CEP4, CEPR1, CEPR2* and *RLK7* (Supplementary Fig. 10). Moreover, *CEP4* promoter activity was not detected in the leaf vasculature (Fig. 4D) but CEP4 induced *FRK1* expression in this tissue (Fig. 1F, Supplementary Fig. 3F). This raises the possibility for CEP4 mobility between different tissue layers and that the peptide may exert its immune-related function in a combination of cell-autonomous and short-to-long distance signaling.

The limited, tissue-specific activity of *CEPR1* and *CEPR2* promoters may also explain the minor contribution of these receptors to mount early CEP4-induced responses upon elicitation in whole seedlings (Fig. 3A–F). For this reason, we generated *CEPR2-GFP* and *CEPR1-GFP* overexpression lines to test whether the constitutive expression of these receptors across tissues can promote CEP4-induced responses. Two *35S::CEPR2-GFP* lines overexpressed the receptor ~100–150 fold compared to Col-0 and protein accumulation was detected by western blots (Supplementary Fig. 11 A, B). We only obtained one *35S::CEPR1-GFP* line with a ~10 fold overexpression relative to Col-0 (Supplementary Fig. 11C). However, we failed to detect CEPR1-GFP accumulation in this line, suggesting that the protein may be unstable. Consistent with a function as a CEP receptor (Fig. 2D–G), the overexpression of CEPR2-GFP enhanced the responsiveness of seedlings to CEP1 and CEP4 in MAPK activation experiments (Fig. 4F). As expected from the lack of detectable protein, *CEPR1* transcript overexpression did not alter CEP4-induced MAPK activation (Supplementary Fig. 11D). However, *CEPR1* overexpression mildly promoted CEP1 and CEP3-induced MPK3/MPK6 phosphorylation (Supplementary Fig. 11D), in line with previous reports of CEPR1 being the primary receptor for canonical class I CEPs[24,27,31,32,44]. This suggests that CEPR2 is a physiologically relevant CEP4 receptor.

## CEPs promote FLS2 signaling and bacterial resistance under reduced nitrogen supply

Unlike other immune-promoting phytocytokines, such as PIPs and SCOOPs, *CEPs* are not strongly transcriptionally regulated upon flg22 perception or *Pto* infection (Supplementary Fig. 1A–C, Fig. 4B). We were thus interested to determine the biological relevance of CEP-mediated control of cell surface immunity. First, we tested whether CEP treatment affects FLS2-dependent signaling. CEP4 application could promote flg22-induced ethylene accumulation and resistance induced by a low dose of flg22 (100 nM) (Fig. 5A, B). Yet, we did not observe a noticeable defect in flg22-induced ethylene accumulation in *cep6x* and *cepr1/2/rlk7*[AEQ] (Supplementary Fig. 12A, B). CEP4 did not induce ethylene production in *cepr1/2/rlk7*[AEQ] (Supplementary Fig. 12B), similar to abolished CEP4-triggered calcium influx and MAPK activation in this mutant background (Fig. 3E, F). Unlike *cepr1-3/2-4* (Supplementary Fig. 7A), the triple receptor mutant did not show elevated basal ethylene levels, raising the question whether RLK7 promotes ethylene accumulation in the absence of CEPR1/CEPR2 (Supplementary Fig. 12B). Also, flg22-induced MAPK activation was unaltered in *cep6x* and *cepr1/2/rlk7*[AEQ], suggesting that early FLS2 signaling is not impaired in seedlings of these genotypes grown under normal growth conditions (Supplementary Fig. 12C, D).

As modulators of systemic N-demand signaling, several *CEPs* are transcriptionally upregulated in N-starved roots, including *CEP1*[24,57,58]. It is known that the N status of plants affects disease resistance to different pathogens, but the underlying molecular mechanisms remain unknown[59,60]. Notably, reduced N availability was shown to promote resistance to *Pto*[61], suggesting a potential link between N homeostasis and antibacterial resistance. This led us to hypothesize that CEPs might be involved in the coordination of the plant's N status with cell surface immunity. To test this, we transferred two-week-old seedlings for 24 h to ½ MS medium with standard (100%: 20 mM $NO_3^-$, 10 mM $NH_4^+$) or reduced N concentrations (10%: 2 mM $NO_3^-$, 1 mM $NH_4^+$; 5%: 1 mM $NO_3^-$, 0.5 mM $NH_4^+$; 1%: 0.2 mM $NO_3^-$, 0.1 mM $NH_4^+$) before challenging them with flg22. Mild reduction in N content (2 mM $NO_3^-$, 1 mM $NH_4^+$; 1 mM $NO_3^-$, 0.5 mM $NH_4^+$) corresponding to 10% and 5% N of standard ½ MS, respectively, promoted flg22-induced MAPK activation, while very low N concentration did not (0.2 mM $NO_3^-$, 0.1 mM $NH_4^+$) (Supplementary Fig. 12E). This suggests that different N concentrations modulate FLS2 signaling capacity. Going forward, we focused on 10% N (2 mM $NO_3^-$, 1 mM $NH_4^+$), since this concentration had the strongest effect on FLS2-mediated MAPK phosphorylation (Supplementary Fig. 12E). Interestingly, the enhancement of flg22-induced MAPK activation upon reduced N treatment was compromised in *cep6x*, suggesting that CEPs promote FLS2 signaling under lower N conditions (Fig. 5C, E, Supplementary Fig. 12F–H). Surprisingly though, the *cep5x* mutant also lost the low N promotional effect on flg22-triggered MAPK activation (Fig. 5C, E). We next tested whether CEP receptors are required for the N-dependent regulation of FLS2 signaling. Indeed, the promotional effect of reduced N condition on flg22-induced MAPK activation was decreased in *cepr1/2/rlk7*[AEQ] (Fig. 5D–F, Supplementary Fig. 12G–I). Consistent with our results comparing *cep5x* and *cep6x* for reduced N-promoted flg22-triggered MAPK activation (Fig. 5C–E), the *cepr1/2*[AEQ] showed a similarly weaker response, suggesting that CEP4-RLK7 function is dispensable in this biological context (Fig. 5D–F). We next examined if reduced N availability can promote bacterial resistance. To test this, we grew seedlings in solid phytagel plates with ½ MS, which allowed us to define N concentrations before infection with *Pto*[COR-] by flood inoculation. As previously reported[61], reduced N generally promoted resistance to *Pto*[COR-] in wild-type seedlings (Fig. 5G, Supplementary

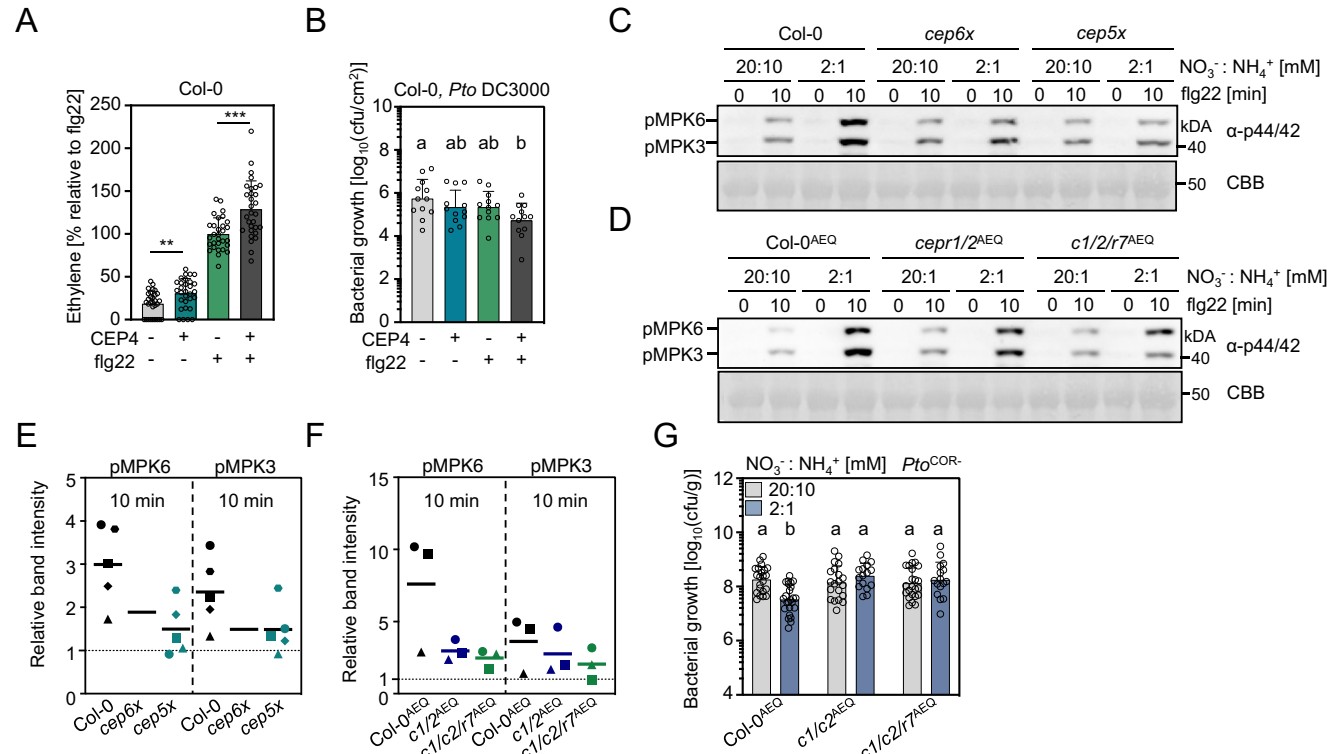

**Fig. 5 | CEPs and CEPRs promote FLS2 signaling and antibacterial resistance under reduced N conditions. A** Ethylene concentration in Col-0 leaf discs 3.5 h after mock (ddH$_2$O), CEP4 (1 μM) and/or flg22 (500 nM) treatment; n = 30, pooled from seven independent experiments with mean ± SD (Mann-Whitney U test, **p = 0.0042; Welch's t-test, ***p = 0.0001). **B** cfu of *Pto* DC3000 in Col-0 leaves 3 dpi following mock (ddH$_2$O), CEP4 (1 μM) and/or flg22 (100 nM) pre-treatment; n = 12 pooled from three independent experiments with mean ± SD (one-way ANOVA, Tukey post hoc, a-b p = 0.0162). **C, D** MAPK activation upon flg22 (100 nM) treatment after 24 h transfer of seedlings to medium containing indicated concentrations of N. Western blots were probed with α-p44/42. Size marker is

indicated. CBB = Coomassie brilliant blue. **E, F** Quantification of pMPK6/pMPK3 band intensities normalized to the CBB band and relative to flg22-treated standard N (20 mM NO$_3$$^-$: 10 mM NH$_4$$^+$) of the respective genotype (set as 1 using ImageJ software). **E** n = 5, **F** n = 3, with mean. Different symbols represent independent experiments. **G** cfu of *Pto*$^{COR-}$ 3 days post flood inoculation. Col-0$^{AEQ}$ (n = 22), c1/2$^{AEQ}$ (n = 20), c1/2/r7$^{AEQ}$ (n = 23) for standard N and Col-0$^{AEQ}$ (n = 24), c1/2$^{AEQ}$ (n = 16), c1/2/r7$^{AEQ}$ (n = 16) for reduced N, pooled from three independent experiments with mean ± SD (one-way ANOVA, Tukey post-hoc test, a-b p < 0.005). All experiments were performed at least three times in independent biological repeats with similar results.

Fig. 12J). This effect was abolished in both *cepr1/2*$^{AEQ}$ and *cepr1/2/rlk7*$^{AEQ}$ (Fig. 5G) and reduced in *cep6x* and *cep5x* mutants (Supplementary Fig. 12J). Surprisingly, under these experimental conditions, the tested mutants did not show enhanced susceptibility with saturated N, which might be related to the specific growth conditions and inoculation technique on phytagel plates required to assess the impact of N availability on disease resistance (Fig. 5G, Supplementary Fig. 12J). Regardless, in line with our flg22-induced MAPK results under different N conditions (Fig. 5C–F), this data reinforces that the canonical CEP-CEPR1/2 signaling pathway, but not CEP4-RLK7, modulates Arabidopsis resistance to *Pto* under nitrogen-limiting conditions.

We raised the question whether the impairment in FLS2 signaling and bacterial resistance under reduced N conditions in our higher-order CEP/CEP receptor mutants might be associated with defects in N homeostasis. The No-0 *cepr1/2* mutant has paler leaves, smaller rosette size, and constitutive anthocyanin accumulation related to defects in nitrate uptake even under N-sufficient growth conditions[24]. Yet, the *cep6x* and *cep5x* shoot is visibly indistinguishable from the wild-type plant (Supplementary Fig. 2D). We observed reduced chlorophyll content in *cepr1-3* in soil-grown plants (Supplementary Fig. 13A). However, chlorophyll contents remained unchanged in *cep6x*, and mutating *cepr2* and *rlk7* in *cepr1* mutant backgrounds did not further enhance this phenotype, indicating that only *cepr1* mutants show possible defects in photosynthetic activity (Supplementary Fig. 13A). This is interesting since under normal

growth conditions, the *cepr1-3* single mutant did not show impairment in resistance to *Pto*$^{COR-}$ (Fig. 2A) indicating that the observed susceptibility phenotype in higher-order receptor and *cep* ligand mutants (Fig. 2A, Supplementary Fig. 4B, 8H) is not a mere consequence of miss-regulated N homeostasis. Additionally, we grew *cep6x* and *cepr1/2/rlk7*$^{AEQ}$ seedlings in standard or reduced N media. We noticed a mild decrease in seedling growth of all genotypes under moderate N supply (2 mM NO$_3$$^-$, 1 mM NH$_4$$^+$), which was more pronounced under low N conditions (0.2 mM NO$_3$$^-$, 0.1 mM NH$_4$$^+$) after seven days of growth (Supplementary Fig. 13B, C). The *cep6x* mutant response was indistinguishable from the WT (Supplementary Fig. 13B), whereas the *cepr1/2/rlk7*$^{AEQ}$ seedlings showed slightly but not significantly reduced growth at lower N concentrations compared to its WT control (Supplementary Fig. 13C). This further supports that impairment in lower N-induced promotion of FLS2 signaling and bacterial resistance cannot be explained by deregulated metabolism in *cep6x* and *cepr1/2/rlk7*$^{AEQ}$.

Strongly reduced nitrate concentrations enhance the expression of the high-affinity nitrate transporter *NRT2.1*[62–64]. We could not detect enhanced *NRT2.1* expression in WT under our reduced N conditions, which remains higher than nitrate concentrations previously tested for *NRT2.1* expression (2 mM nitrate vs 1 mM) (Supplementary Fig. 13D)[64]. However, flg22 promoted *NRT2.1* transcript accumulation which was abrogated in *cep6x* (Supplementary Fig. 13D). Together, these results indicate a CEP-and CEP receptor-dependent connection between FLS2-triggered PTI, bacterial resistance and the plant´s N supply, revealing a

previously unknown mechanism of signaling cross-talk between cell surface immunity and the plant's nutritional status.

## Discussion

Recently, phytocytokines received increasing attention due to their pivotal role in mediating plant responses to environmental challenges and biotic stress[8,22,25,47,65]. Unlike classical phytohormones, these peptides are derived from a precursor protein encoded by a single gene, allowing for rapid synthesis and activation without complex metabolic pathways[66]. This simplicity has enabled fast evolution and efficient regulation, providing plants with diverse signaling options for high adaptability under changing conditions[66]. Despite the redundancy within phytocytokine multigene families, new functions of endogenous peptides are continuously being discovered. CEPs, for example, are known to influence growth and development, particularly by modulating root system architecture and nutrient uptake through CEPR1 and CEPR2[24,26–32]. Our study shows that CEP signaling is also important for controlling immunity in Arabidopsis. In addition to canonical CEPRs, we identified that PTI-related CEP signaling also involves the CEPR-related RLK7. Together, these three partially cell-type and tissue-specific receptors play a vital role in regulating plant immunity and likely coordinate biotic stress responses with nutritional cues, thereby enhancing the plant's overall adaptability.

CEP4 triggers hallmark PTI responses and directly binds to both CEPR2 and RLK7, and CEP4 immune-related outputs show variable grades of dependency on all three CEP receptors, including the initially identified CEP binding receptors CEPR1 and CEPR2. The canonical group I CEP1 and CEP9.5, however, exclusively depend on CEPR1/2 and CEP1 does not bind to RLK7. CEPR2's contribution to CEP signaling was unclear based on its marginal involvement in CEP1-mediated N demand signaling[24]. Here, we show that CEPR2 binds CEP4, and *CEPR2* overexpression promotes both CEP4 and canonical CEP1-mediated MAPK activation, reinforcing its function as a bona fide CEP receptor. In addition to CEP4, RLK7 also recognizes closely-related PIP peptides to regulate growth, salt stress, and immunity, and binds PIP1 with higher affinity than CEP4, suggesting preferential recognition of this ligand[25,53,55] (Fig. 3J).

CEP orthologues can be found in all seed plants, indicating an evolutionarily older signaling pathway compared to PIP and PIP-LIKEs (PIPL), which are only predicted to be encoded in angiosperm genomes[35,42,66]. Active domains of both families are comparable in size, share the GxGH motif located at the C-terminus of the functional peptide, and, their bioactivity is strongly affected by proline hydroxylation[34,55,67]. Immunomodulatory PIP1 has two key proline residues[55], with the first shared with CEP4 and the second conserved in canonical CEPs, suggesting that CEP4's receptor-specificities are unique among CEPs, likely caused by its distinct sequence (Supplementary Fig. 1D). Hydroxylation of the first proline residue is critical for PIP1's bioactivity[55], raising the question if it might facilitate CEP4-RLK7 interactions. It will be interesting to compare the molecular mechanisms of PIP1/CEP4-RLK7 and CEP1/CEP4-CEPR1/CEPR2 recognition and activation in future structure analyses and examine receptor dependency of the other group I CEPs.

It is not unusual that members of multigene peptide families are recognized by closely related or partially redundant receptors with varying quantitative contributions. Members of the CLE peptide family also bind multiple receptors with distinct affinities to regulate epidermal cell patterning[46,68], and PEPR1 and PEPR2's contribution to PEP perception shows ligand-dependent differences[69–72]. Similarly, CEP/PIP/PIPL-related INFLORESCENCE DEFICIENT IN ABCISSION (IDA) signaling, although primarily dependent on HAESA (HAE) and HAESA-LIKE 2 (HSL2), is not fully abolished in the *hae hsl2* double mutant, suggesting an involvement of other RLKs in its perception[67,73].

The overlap of CEP4 with RLK7 expression in leaves but not with *CEPR1* and *CEPR2*, suggests a combination of cell-autonomous and short-distance signaling contributing to CEP-mediated immune modulation in the shoot. This is in line with previous results, showing that the canonical CEP-CEPR1/2 signaling can function through short- and long-distance pathways[24,74]. We detected other CEP transcripts in the shoot, and our results with the *cep5x* mutant revealed that the canonical group I CEPs also contribute to resistance against *Pto*COR-. A challenge for the future will be to determine how three receptors with distinct expression patterns integrate responses between tissues and the concerted action of multiple ligands. Promoter activity studies are valuable for gaining insights into tissue specificity, but they can be influenced by transgene insertion sites and the presence or absence of specific enhancer/silencer sequences. Dissecting spatiotemporal ligand availability and CEP-CEPR1/CEPR2/RLK7 signaling specificity will be an important future task.

CEPs promote FLS2 signaling and antibacterial resistance under reduced N conditions, but the mechanistic basis remains unknown. Other phytocytokines modulate defence signaling by regulating PRR abundance or receptor complex formation and dynamics[8,9,11,12]. FLS2 is expressed in multiple tissues, including the vasculature, stomata and the epidermis[75] and thus could be directly or indirectly regulated by tissue-specific CEP receptors. It will be interesting to reveal whether CEPs directly or indirectly modulate FLS2 signaling and whether this might translate to other PRRs.

Our data suggest that a reduction in N supplementation promotes FLS2 activation and resistance against *Pto* in a CEP and CEPR1/CEPR2-dependent manner. While CEP4-RLK7 signaling contributes to disease resistance in soil-grown plants (Supplementary Fig. 4B, C, Fig. 3M, Supplementary Fig. 8H), its function in promoting PTI under lower N conditions is negligible. This is consistent with the described role of the canonical CEP-CEPR1/2 pathway in N signaling and suggests a minor role for CEP4-RLK7 or PIP-RLK7 in this biological context[24]. Accumulating evidence suggests a direct integration of nutrient homeostasis and PTI in plants. Perception of flg22 by FLS2 induces PHT1.4 phosphorylation to inhibit phosphate (Pi) uptake and promote root immunity[76]. Similarly, a recent study revealed a cross-talk between PTI and nutrition by showing that under low iron (Fe) conditions the flg22-FLS2 signaling module suppresses Fe uptake through a localized degradation of the iron uptake-regulating Iron Man 1 (IMA1)[77]. It remains unknown whether immune activation also regulates N transport in root or shoot tissue. CEPs induce nitrate, Pi and sulfate uptake, suggesting CEP-CEPR1/CEPR2/RLK7-dependent modulation of several transporter pathways[78]. This raises the question whether nutrient uptake directly contributes to CEP-mediated immune modulation. The main source of inorganic N for plant utilization is nitrate, which also functions as a signaling molecule to induce adaptive growth responses[79]. Nitrate is sensed by the plasma membrane transceptor NRT1.1 and the nuclear transcriptional regulator NLP7[80,81]. NRT1.1 and similar transporters are regulated by phosphorylation to control transport activity, including NRT1.2 phosphorylation by CEPR2[81–86], which we identified as a CEP4 receptor. It will be interesting to resolve whether and how cell surface signaling and nitrogen sensing/transport directly or indirectly intersect.

Since seedlings grown under high N concentrations (as provided by standard ½ MS medium) show limited flg22 responsiveness and bacterial resistance, it is also possible that N saturation inhibits PTI by suppressing *CEP* expression and accumulation. Indeed, in Medicago *CEP1* expression is negatively regulated by direct binding of NLP1 to the CEP1 promoter in response to high nitrate levels[87]. Additionally, in line with this hypothesis, lower nutrient conditions were shown to enhance flg22-dependent PTI in gnotobiotically grown plants, which was promoted by microbial colonization[88]. This effect was suppressed when a standard concentration of N was resupplied, suggesting that high N levels can have a negative impact on immune outputs, but the underlying molecular mechanism remains unknown[88]. It will be interesting to test whether N and CEP-dependent PTI modulation is

similarly influenced by the plant's microbiome in Arabidopsis, especially considering CEP-CRA2's role in supporting symbiotic rhizobia in Medicago[89–91].

CEPR1/CEPR2/RLK7 and CEPs are widely conserved among angiosperms, including economically relevant crop plants[42]. CEPs are important to promote nodulation in legumes and also regulate sucrose-dependent root growth inhibition and fecundity in Arabidopsis[28,43,91,92]. This places CEPs as central integrators of biotic interactions (symbiosis and pathogen defence) with plant nutrition, growth and development[30,78]. It will be critical to understand whether N-dependent and CEP-mediated PTI modulation extends beyond Arabidopsis, and to decipher how CEPs can promote immunity and control symbiosis in diverse species. This will provide important insights for future crop improvement strategies that coordinate crop nutrition with disease resistance.

## Methods

### Molecular cloning
To generate *CEP4* overexpression lines, the coding sequence of *CEP4* (AT2G35612) was synthesized (Twist Bioscience, USA) with attB attachment sites for subsequent gateway cloning into pDONRZeo (Invitrogen, USA) and recombination with pB7WG2 (VIB, Ghent). To generate CRISPR-Cas9 mutants, appropriate target sites (two per gene of interest) were designed using the software tool chopchop (https://chopchop.cbu.uib.no/). Individual guide RNA constructs containing gene-specific target sites were synthesized (Twist Bioscience, USA) and subsequently stacked in a GoldenGate-adapted pUC18-based vector. To generate different order CRISPR *cep* mutants, *cepr1/2*[AEQ], *rlk7/iku2*[AEQ] and *rlk7*[AEQ] 12, 4, 4 and 2 target site-containing gRNA constructs were stacked, respectively (Supplementary Table 2). Together with FastRed-pRPS5::Cas9, higher-order gRNA stacks were subsequently cloned into pICSL4723 for in planta expression[93].

To generate the *pCEP4::NLS-3xmVenus*, *pCEPR1::NLS-3xmVenus* and *pCEPR2::NLS-3xmVenus* reporter constructs, 1000, 1696 and 2788 bp fragments upstream of the start codon, respectively, were amplified from genomic DNA and assembled together with the sequence coding for the nuclear localization signal of SV40 large T antigen followed by 3 consecutive mVenus YFP fluorophores[94] into a GoldenGate-modified pCB302 binary vector for plant expression. For *pRLK7* the 1957 bp promoter sequence upstream from the start codon was amplified with primers containing attB attachment sites for subsequent gateway cloning into pDONRZeo (Invitrogen, USA) and recombination with promotor::Venus (YFP)-H2B destination vector[94]. For *CEPR1* (AT5G49660) and *CEPR2* (AT1G72180) overexpression lines, the coding sequence of both genes was amplified from cDNA with attB attachment sites for subsequent cloning into a pDONR223 (Invitrogen, USA) and recombination with pK7FWG2 (VIB Ghent, Belgium). All of the generated plant expression constructs were subsequently transformed into *Agrobacterium tumefaciens* strain GV3101 before floral dip transformation of Arabidopsis. All primers used for cloning are listed in (Supplementary Table 3).

### Plant material and growth conditions
Arabidopsis Col-0, Col-0[AEQ][39] and No-0 were used as wild types for experiments and generation of transgenic lines or CRISPR mutants. The *cepr1-3* (GK-467C01), *cepr2-3* (SALK_014533), *rlk7-1* (SALK_056583), *rlk7-3* (SALK_120595), *iku2-4* (Salk_073260) and the novel *cepr2-4* allele (GK-695D11) were obtained from NASC (UK)[28,55,74,95]. The No-0 *cepr1-1xcepr2-1* was obtained from RIKEN (Japan)[24]. T-DNA insertion mutants were genotyped by PCR using T-DNA- and gene-specific primers as listed in Supplementary Table 3. The lack of *CEPR2* transcript in *cepr2-4* (GK-695D11) was determined by semi-quantitative PCR from cDNA using *cepr2-4* genotyping primers (Supplementary Fig. 5A). The *cepr1-3/2-4* double mutant was obtained by genetic crossing. The *bak1-5/bkk1-1* mutant was characterized previously[38]. For visualizing tissue-

specific *FRK1* expression, a *pFRK1::NLS-3xmVenus* line was used[41]. To isolate homozygous CRISPR mutants, pICSL4723 transformant T1 seeds showing red fluorescence were selected and grown on soil before genotyping with gene-specific primers and Sanger sequencing (Supplementary Table 3). Mutants lacking the transgene were identified by loss of fluorescence. To generate the *cep6x 35S::CEP4* lines, the same pB7WG2 CEP4 construct used for the generation of *CEP4* overexpression lines was utilized for floral dip transformation of homozygous *cep6x*.

Plants for physiological assays involving mature plants were vernalized for 2-3 days in the dark at 4 °C and later grown in individual pots in environmentally controlled growth rooms (20-21 °C, 55% relative humidity, 8 h photoperiod). For seedling-based assays, seeds were sterilized using chlorine gas and grown axenically on ½ Murashige and Skoog (MS) media supplemented with vitamins (Duchefa, Netherlands), 1% sucrose, with or without 0.8% agarose at 22 °C and a 16 h photoperiod unless stated otherwise. For experiments using ½ MS medium with reduced N concentrations, modified MS salts without nitrogen-containing compounds (Duchefa, Netherlands) were used and supplemented with $KNO_3/NH_4NO_3$ to achieve 100% N ($KNO_3$ 9.395 mM, $NH_4NO_3$ 10.305 mM), 10% N ($KNO_3$ 0.9395 mM, $NH_4NO_3$ 1.0305 mM), 5% ($KNO_3$ 0.4698 mM, $NH_4NO_3$ 0.5153 mM) and 1% ($KNO_3$ 0.09395 mM, $NH_4NO_3$ 0.10305 mM) conditions. To keep the ionic strength equal in 10%, 5% and 1% N conditions, media were supplemented with 90% (8.455 mM), 95% (8.925 mM) and 99% (9.301 mM) KCl, respectively.

### Grafting
Arabidopsis seedlings were grown vertically on ½ MS agar medium without sucrose in short-day conditions seven days before grafting. Grafting was performed aseptically under a stereo microscope as previously described[96]. Vertically mounted plates with reciprocally grafted seedlings were returned to short-day conditions for 10 days. Healthy seedlings were transferred to the soil.

### Imaging and microscopy
Confocal laser-scanning microscopy was performed using a Leica TCS SP5 (Leica, Germany) microscope (with Leica Application Suite X 3.7.4.23463). For the mVenus fluorophore, pictures were imaged with argon laser excitation at 514 nm and a detection window of 525–535 nm. Propidium iodide was visualized using DPSS 561 laser emitting at 561 nm with a detection window of 610–630 nm. For analyzing the promoter activity of untreated NLS-3xmVenus or Venus (YFP)-H2B reporter lines under the control of different promoters (*pCEP4*, *pCEPR1*, *pCEPR2* and *pRLK7*), 7- (for roots) or 12-day (for shoots) old vertically-grown seedlings were stained with propidium iodide immediately before microscopic analysis. The laser power used to excite the fluorophores was adjusted according to the intensity of the fluorescence emitted. As a result, the laser power used for excitation varied across different reporter lines depending on the activity level of the tested promoters. Similarly, the Z-stack step size varied between reporter lines due to a different tissue-specific expression pattern of each promoter and was adjusted to capture all the cell layers where the signal was active. For imagining promoter activity after treatment, 12-day-old seedlings were transferred to a 24-well plate containing $ddH_2O$ (mock) or indicated concentration of peptide solution. For comparison purposes, seedlings of the same genotype were imaged using identical laser intensities and interval/number of slices for Z stack projection 16 h after treatment.

### Calcium influx assay
Apoaequorin-expressing liquid-grown eight-day-old seedlings were transferred individually to a 96-well plate containing 100 μl of 5 μM coelenterazine-h (PJK Biotech, Germany) and incubated in the dark overnight. Luminescence was measured using a plate reader

(Luminoskan Ascent 2.1, Thermo Fisher Scientific, USA). Background luminescence was recorded by scanning each well 12 times at 10 s intervals, before adding a 25 µl elicitor solution to the indicated final concentration. Luminescence was recorded for 30 min at the same interval. The remaining aequorin was discharged using 2 M $CaCl_2$, 20% ethanol. The values for cytosolic $Ca^{2+}$ concentrations ($[Ca^{2+}]_{cyt}$) were calculated as luminescence counts per second relative to total luminescence counts remaining ($L/L_{max}$).

## Ethylene measurement
Leaf discs (4 mm in diameter) from four-to-five-week-old soil-grown Arabidopsis were recovered overnight in $ddH_2O$. Three leaf discs per sample were transferred to a glass vial containing 500 µl of $ddH_2O$ before adding $ddH_2O$ (mock) or peptides to the indicated final concentration. Glass vials were capped with a rubber lid and incubated under gentle agitation for 3.5 h. One mL of the vial headspace was extracted with a syringe and injected into a Varian 3300 gas chromatograph (Varian, USA) to measure ethylene.

## MAPK activation and western blot analysis
Five-day old Arabidopsis seedlings growing on ½ MS agar plates, were transferred into a 24-well plate containing liquid medium for seven days. 24 h before the experiment, seedlings were equilibrated in a fresh ½ MS medium. For N reduction experiments, modified ½ MS containing 100% N, 10% N, 5% N and 1% N supplemented with KCl was used. MAPK activation was elicited by adding the peptides to the indicated concentrations. Six seedlings per sample were harvested, frozen in liquid nitrogen and homogenized using a tissue lyser (Qiagen, Germany). Proteins were extracted using a buffer containing 50 mM Tris-HCl (pH 7.5), 50 mM NaCl, 10% glycerol, 5 mM DTT, 1% protease inhibitor cocktail, 1 mM phenylmethylsulfonyl fluoride (PMSF), 1% IGEPAL, 10 mM EGTA, 2 mM NaF, 2 mM $Na_3VO_4$, 2 mM $Na_2MoO_4$, 15 mM ß-Glycerophosphate and 15 mM p-nitrophenylphosphate before analysis by SDS-PAGE and western blot. Phosphorylated MAPKs were detected by α-p44/42 antibodies (Cell Signaling, USA). To quantify the intensity of the specific bands, ImageJ software (version 1.53t) was used. Each band was selected with the same-sized frame and the intensity peak was determined. The area under each peak was calculated and normalized to Coomassie staining as a measure of relative band intensity (RBI). The RBI of each genotype at 100% N upon flg22 treatment was set to one.

To determine CEPR1 and CEPR2 protein levels in *35S::CEPR1-GFP* and *35S::CEPR2-GFP* overexpression lines, seedlings were grown in ½ MS liquid medium for 12 days. Afterwards, harvested seedlings were frozen in liquid nitrogen, homogenized using a tissue lyser (Qiagen, Germany) and the proteins were isolated using an extraction buffer contenting 50 mM Tris-HCl (pH 7.5), 50 mM NaCl, 10% glycerol, 2 mM EDTA, 2 mM DTT, 1% protease inhibitor cocktail, 1 mM phenylmethylsulfonyl fluoride (PMSF) and 1% IGEPAL. After SDS-PAGE and western blot, GFP-tagged proteins were detected by α-GFP antibodies (ChromoTek). Uncropped blots are shown in Source Data file.

## Seedling growth inhibition
Arabidopsis seedlings were grown for five days on ½ MS agar plates before the transfer of individual seedlings into each well of a 48-well plate containing liquid medium with or without elicitors in the indicated concentration. After seven-day treatment, the fresh weight of individual seedlings was measured.

## Pathogen growth assay
*Pseudomonas syringae* pv. *tomato* (*Pto*) DC3000 and *Pto* lacking the effector molecular coronatine $Pto^{COR-}$ were grown on King's B agar plates containing 50 µg/mL rifampicin and 50 µg/mL kanamycin at 28 °C. After two-to-three days bacteria were resuspended in $ddH_2O$

containing 0.04% Silwet L77 (Sigma Aldrich, USA). The bacterial suspension was adjusted to an $OD_{600} = 0.2$ ($2 \times 10^8$ cfu/mL) for $Pto^{COR-}$ or $OD_{600} = 0.02$ ($2 \times 10^7$ cfu/mL) for *Pto* DC3000 before spray inoculating four-to-five-week-old plants. For peptide-induced resistance in local tissues, $ddH_2O$ (mock), flg22 (100 nM) and/or CEP4 (1 µM) were syringe-infiltrated into mature leaves. After 24 h, *Pto* DC3000 ($OD_{600} = 0.0002$, $2 \times 10^5$ cfu/mL) was syringe-infiltrated into pre-treated leaves and incubated for three days before determining bacterial counts. For CEP4-induced resistance in systemic tissues, CEP4 (1 or 5 µM) or $ddH_2O$ (mock) were syringe-infiltrated into the first two true leaves of young three-to-four-week-old Arabidopsis. After four days, *Pto* DC3000 ($OD_{600} = 0.0002$, $2 \times 10^5$ cfu/mL) was syringe-infiltrated into leaves three and four of the pre-treated plants. Bacterial counts were determined four days after infection.

## Systemic acquired resistance
SAR experiments were performed as previously described[97]. Briefly, plants were cultivated in a mixture of substrate (Floradur) and silica sand in a 5:1 ratio under short day (SD) conditions (10 h) in a growth chamber at 22 °C /18 °C (day/night) with a light intensity of 100 µmol $m^{-2}s^{-1}$, and 70% relative humidity (RH). SAR assays were performed using *Pto* DC3000 and *Pto* AvrRpm1. Bacteria were grown on NYGA media (0.5% peptone, 0.3% yeast extract, 2% glycerol, 1.8% agar, 50 µg/mL kanamycin, 50 µg/mL Rifampicin) at 28 °C before infiltration. Freshly grown *Pto* avrRpm1 was diluted in 10 mM $MgCl_2$ (to reach a final concentration of $1 \times 10^6$ cfu/mL) and syringe-infiltrated in the first two true leaves of four-and-a-half-week-old plants. Concurrently, 10 mM $MgCl_2$ was applied to a separate set of plants as the mock control treatment. Three days after *Pto* avrRpm1 infiltration, plants were challenged in their 3rd and 4th leaves with *Pto* DC3000 ($1 \times 10^5$ cfu/mL). Bacterial titers were determined four days after *Pto* DC3000 infection.

## Flood inoculation
To test for N-depended bacterial resistance, Arabidopsis seedlings were grown under long-day conditions (16 h photoperiod, 22 °C) on sucrose-free ½ MS plates solidified with 0.9% phytagel (Sigma Aldrich, USA) and supplemented with either 100% N ($KNO_3$ 9.395 mM, $NH_4NO_3$ 10.305 mM) or 10% N ($KNO_3$ 0.9395 mM, $NH_4NO_3$ 1.0305 mM, KCl 8.455 mM). Approximately two weeks post-germination, the axenically-grown seedlings were flood-inoculated[98] with a bacterial suspension of $Pto^{COR-}$ dissolved in sterile $ddH_2O$ containing 0.025% Silwet L-77 at $OD_{600} = 0.001$ ($10^6$ cfu/mL). After three minutes, the bacterial suspension was removed, and the plates were sealed with Micropore tape. Three days later, the individual shoots were harvested, sterilized three times with 5% $H_2O_2$, and washed thoroughly with sterile $ddH_2O$ water. Each surface-sterilized seedling was ground twice in 200 µL of $ddH_2O$ with tissue lyser (2.5 min, 25 Hz). To examine bacterial growth, serially diluted samples were plated on an LB medium containing 50 mg/L rifampicin. Two days after plating, CFUs were counted and normalized as CFU/g fresh weight.

## Gene expression analysis
For seedlings-based assays, 12-day-old liquid-grown seedlings were equilibrated in fresh medium for 24 h before treatment with the indicated peptides. For adult plants, four-to-five-week-old Arabidopsis leaves were syringe-infiltrated with $ddH_2O$ (mock), flg22 (1 µM) or *Pto* DC3000 ($OD_{600} = 0.001$, $5 \times 10^5$ cfu/mL) and incubated for 24 h. All samples for RT-qPCR analysis were harvested at the indicated time points, frozen in liquid nitrogen and homogenized using a tissue lyser (Qiagen, Germany). Total RNA was isolated using TRIzol reagent (Roche, Switzerland) and purified using Direct-zol™ RNA Miniprep Plus kit (Zymo Research, Germany). 2 µg of the total RNA was digested with DNase I and reverse transcribed with oligo (dT)18 and Revert Aid

reverse transcriptase. RT-qPCR experiments were performed using Takyon™ Low ROX SYBR MasterMix (Eurogentec, Belgium) with the AriaMx Real-Time PCR system (Agilent Technologies, USA). Expression levels of all tested genes were normalized to the house-keeping gene *Ubiquitin 5* (*UBQ5*). Sequences of all primers used for RT-qPCR analysis are found in Supplementary Table 3.

### Expression and purification of recombinant receptor ectodomains

*Spodoptera frugiperda* codon-optimized synthetic genes (Invitrogen GeneArt), coding for Arabidopsis CEPR1 (residues 23 to 592), CEPR2 (residues 32 to 620) and RLK7 (residues 29 to 608) were cloned into a modified pFastBAC vector (Geneva Biotech) providing a 30 K signal peptide[99], a C-terminal TEV (tobacco etch virus protease) cleavable site and a StrepII-9xHis affinity tag. For protein expression, *Trichoplusia ni Tnao38* cells[100] were infected with CEPR1, CEPR2 or RLK7 virus with a multiplicity of infection (MOI) of 3 and incubated one day at 28 °C and two days at 21 °C at 110 rounds per minute (rpm). The secreted proteins were purified by $Ni^{2+}$ (HisTrap excel, Cytiva, equilibrated in 25 mM $KP_i$ pH 7.8 and 500 mM NaCl) followed by Strep (Strep-Tactin Super-flow high-capacity, IBA Lifesciences, equilibrated in 25 mM Tris pH 8.0, 250 mM NaCl, 1 mM EDTA) affinity chromatography. All proteins were incubated with TEV protease to remove the tags. Proteins were purified by SEC on a Superdex 200 Increase 10/300 GL column (Cytiva, USA) equilibrated in 20 mM citrate pH 5.0, 150 mM NaCl and further concentrated using Amicon Ultra concentrators from Millipore (Merck, Germany) with a 30,000 Da molecular weight cut-off. Purity and structural integrity of the different proteins were assessed by SDS-PAGE.

### Relative chlorophyll content
The relative leaf chlorophyll content in 5-week-old Arabidopsis plants was measured using a portable chlorophyll meter (SPAD-502; Minolta, Tokyo, Japan). Three measurements per leaf were taken, and values were averaged and quantified in SPAD (Soil Plant Analysis Development) units, correlating to chlorophyll content[101,102].

### Analytical size-exclusion (SEC) chromatography
Analytical SEC experiments were performed using a Superdex 200 Increase 10/300 GL column (GE, USA). The columns were pre-equilibrated in 20 mM citric acid pH 5, 150 mM NaCl. 150 µg of CEPR1, CEPR2 and RLK7 were injected sequentially onto the column and eluted at 0.5 mL/min. Ultraviolet absorbance (UV) at 280 nm was used to monitor the elution of the proteins. The peak fractions were analyzed by SDS-PAGE followed by Coomassie blue staining.

### Isothermal titration calorimetry (ITC)
Experiments were performed at 25 °C on a MicroCal PEAQ-ITC (Malvern Instruments, UK) using a 200 µL standard cell and a 40 µL titration syringe. CEP1, CEP4, CEP4[scr] and PIP1 peptides were dissolved in the SEC buffer to match the receptor protein. A typical experiment consisted of injecting 1 µL of a 300 µM solution of the peptide into 30 µM CEPR2 or RLK7 solution in the cell at 150 s intervals. ITC data were corrected for the heat of dilution by subtracting the mixing enthalpies for titrant solution injections into protein-free ITC buffer. Experiments were done in duplicates and data were analyzed using the MicroCal PEAQ-ITC Analysis Software provided by the manufacturer. The N values were fitted to 1 in the analysis.

### Synthetic peptides
The flg22 peptide was kindly provided by Dr. Justin Lee (IPB Halle). Other peptides were synthesized by Pepmic (China) with at least 90% purity and dissolved in $ddH_2O$. Sequences of all the synthetic peptides can be found in Supplementary Table 1.

### Statistics and reproducibility
Statistical analyses were performed using GraphPad Prism (Version 10.2.3). Sample size, *p*-values and statistical methods employed are described in the respective figure legends or in the Source data. Differences were considered to be significant at $p < 0.05$.

### Reporting summary
Further information on research design is available in the Nature Portfolio Reporting Summary linked to this article.

## Data availability
All data generated in this study are available in the main text or the Supplementary Information. All newly generated mutant lines are available upon request to M.S. Source data are provided with this paper.

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

## Acknowledgements

We thank Stefanie Ranf for providing GoldenGate vectors for molecular cloning, Ulrich Hammes for advice in grafting experiments, and Mark Youles (TSL Norwich) and Laurence Tomlinson (TSL Norwich) for providing the pICSL4723OD vector for CRISPR-Cas9 cloning. This work was funded by the following agencies: Deutsche Forschungsgemeinschaft (DFG) grants STE2448/3-1, 3-2 (M.S. and J.R.), STE2448/4-1, TRR356 TP B09 (M.S. and H.L.) and SFB924 TP B06 (A.C.V.), the Technical University of Munich (M.S., J.R., R.H., Z.C., C.W. and J.M.), the Research Council of Norway, Grant 230849 (VOL), Australian Research Council, Grant DP200101885 (M.A.D.), the University of Lausanne (C.B.), the Swiss National Science Foundation grants no. 310030_204526 and the European Research Council (ERC) grant agreement no. 716358 (J.S., H.K.L.).

## Author contributions

Conceptualization: M.S.; Investigation: J.R., H.L., H.K.L., C.B., S.N., C.W., J.M., Z.C., V.O.L., M.S.; Funding acquisition: M.S., J.S., R.H., A.C.V., M.A.D.; Project administration: M.S.; Supervision: M.S., J.R., A.C.V., J.S., M.A.D., R.H.; Writing – original draft: M.S., J.R.; Writing – review & editing: H.L., H.K.L., C.B., S.N., C.W., J.M., V.O.L., M.A.D., A.C.V., R.H., J.S.

## Funding

## Competing interests

The authors declare no competing interests.
