## [Peer Review File · Nature Communications]

REVIEWER COMMENTS

Reviewer #2 (Remarks to the Author):

The authors performed additional experiments in accordance with the referee comments and have added many additional data in the revised manuscript. This manuscript demonstrates that CEP4 binds to RLK7 in addition to CEPR1/2 and activates the PIP pathway. Unlike systemic nitrogen demand signaling, it is surprising but interesting that CEP expressed in leaves enhances immunity by being recognized by receptors in leaves.

However, despite the increase in data, the manuscript has become very difficult to understand because the genes observed and the mutants used vary from experiment to experiment. In particular, in the section discussing the involvement of the CEP pathway in immunity, the mutants used differ from experiment to experiment, hampering the reader's understanding (see attached table). I feel that the authors have arbitrarily adopted only convenient data. Without constantly comparing the responses of the four mutants *cep6x*, *cep5x*, *cepr1/2*, and *cepr1/2/rlk7*, it is not clear whether the observed responses are mediated through the CEP4-RLK7 pathway, the CEP-CEPR pathway, or both. In experiments using only *cep6x* and *cepr1/2/rlk7*, it is not possible to distinguish whether it is the RLK7 pathway or the CEPR pathway that is activated. This is especially important in the experiment in Fig. 5. If CEP4 induction and activation of the RLK7 pathway under low nitrogen is sufficient for upregulation of flg22-induced MPK expression under low nitrogen, then the claim that the CEP family peptides regulate immunity under low nitrogen is misleading to readers. In that case, it should be rephrased as CEP4-RLK7 regulates immunity under low nitrogen. To make the whole story clear, please provide data for the question mark in the table attached, even if it is a negative result.

	Pto COR-growth	Pto DC3000 growth	flg22-induced MPK expression at low N	flg22-induced FRK1 expression at low N
cep6x	Increased	Increased	Decreased	Decreased
cep5x (-CEP4)	Increased	?	?	?
cepr1/2	Increased	Increased	?	?
cepr1/2/rlk7	Increased	More increased	Decreased	?
	Fig. 1I-J Fig. 2A Fig. S4B Fig. S8H	Fig. 3M Fig. S4A	Fig. 5D	Fig. 5F

In addition, to complement the claim that CEP enhances immunity under low nitrogen, I would like to see the sensitivity of four mutants, *cep6x*, *cep5x*, *cepr1/2*, and *cepr1/2/rlk7*, to Pto COR- or Pto DC3000 under low nitrogen.

Fig. 5C is not explained in the text, but is it the roots or the leaves where the expression level of CEP4 was quantified? If it is the roots, the authors should retest the f1g22-dependent MPK expression on the cep6x plants as rootstock and as grafted scion under low nitrogen conditions. For Fig. 5D, the expressions 100% N and 10% N are confusing, please use something like XX mM N.

Comments on the response to Reviewer #1:

The authors have generally answered referee #1's comments accurately.

However, in some experiments such as Fig. 5, immune response is only shown for a few selected mutants. To conclude that "CEPs and their receptors promote immunity in an N status-dependent manner", it is necessary to examine the response for all mutants (*cep6x*, *cep5x*, *cepr1/2*, and *cepr1/2/rlk7*) in each experiment (see also my comment on this point). Experiments using only *cep6x* and *cepr1/2/rlk7* cannot distinguish whether it is the RLK7 or CEPR pathway that is activated under given conditions.

Reviewer #4 (Remarks to the Author):

Phytocytokines are hypothesized to be endogenous peptides that play a pivotal role in integrating diverse external and internal cues to enhance plant health. Among these, C-terminally encoded peptides (CEPs) constitute a class of endogenous peptides known to influence plant growth and participate in the response to nitrogen starvation. However, they have not yet been characterized as phytocytokines involved in plant immunity. In this manuscript, Rzemieniewski et al identified the CEP4 as a novel phytocytokine, recognized by CEP receptors (CEPR1, CEPR2 and RLK7), thereby coordinating plant immunity with nitrogen status. This study proposes that CEPs and their receptors serve as central regulators in adapting biotic stress responses to plant-available resources. Following a round of peer review, the manuscript has undergone improvements in response to reviewer feedback. Nevertheless, certain critical issues raised by reviewers have not been adequately addressed. Addressing these critical issues is crucial for bolstering the reliability of experimental results and amplifying the innovation within the manuscript. These include the "definition issue regarding CEPs as phytocytokine," "whether there are changes in the tissue localization of CEP4, CEPR1, CEPR2, and RLK7 after flg22 treatment or DC3000 infiltration, and the tissue co-localization issue about CEPR receptors with CEP4," "the promotion of immune response under N starvation conditions is mediated by RLK7 or CEPR" and "the elucidation of mechanistic processes by which CEPs may participate in the crosstalk between low-N and PTI, which constitutes the major highlight of the manuscript."

Major:

1. Tissue Expression and Mobility of CEP4:

CEP4 is induced by low nitrogen and expressed in the lateral root emergence zone and hydathode region, while CEPR1 and CEPR2 show expression in vasculature and guard cells, respectively. The authors are urged to provide evidence supporting the mobility of CEP4. Moreover, after treatment with f1g22 or infection with DC3000, clarification is needed on whether there are changes in the tissue expression of CEP4. Additionally, is there any co-expression of CEPR1, CEPR2, or RLK7 with CEP4 at the tissue level post these treatments?

2. Quantitative Analysis of CEP Family Members:

Why weren't all members of the CEP family subjected to quantitative analysis following f1g22 treatment or DC3000 infection, particularly CEP12, which shares the highest homology with CEP4? The authors are strongly encouraged to conduct quantitative analyses on all CEP family members in both roots and shoot post these treatments to establish the correlation between tissue-specific expression patterns and their respective functions.

3. Morphology Phenotypic Assessment Under Low Nitrogen Conditions:

Considering that CEP signaling orchestrates both plant immunity and nitrogen status, there is a need for an elucidation of the morphology phenotypic outcomes resulting from DC3000 inoculation or f1g22 treatment in WT, CEP4-OE, and cep4 mutant plants under low nitrogen conditions.

Minor:

1. The study lacks biochemical evidence supporting the interaction between CEPR1 and CEP4, thereby necessitating additional support for the conclusion that CEPR1 functions as a receptor for CEP4.

2. The authors may provide an explanation of why RLK7 plays a predominant role in the full CEP4 response from an evolutionary perspective (Lines 217-218).

3. In Lines 45-46, it is advisable to include literature citations introducing the concept of phytoytokines.

4. The result description in Lines 85-86 should incorporate corresponding figures.

REVIEWER COMMENTS

Reviewer #2 (Remarks to the Author):

The authors performed additional experiments in accordance with the referee comments and have added many additional data in the revised manuscript. This manuscript demonstrates that CEP4 binds to RLK7 in addition to CEPR1/2 and activates the PIP pathway. Unlike systemic nitrogen demand signaling, it is surprising but interesting that CEP expressed in leaves enhances immunity by being recognized by receptors in leaves.

However, despite the increase in data, the manuscript has become very difficult to understand because the genes observed and the mutants used vary from experiment to experiment. In particular, in the section discussing the involvement of the CEP pathway in immunity, the mutants used differ from experiment to experiment, hampering the reader's understanding (see attached table). I feel that the authors have arbitrarily adopted only convenient data. Without constantly comparing the responses of the four mutants *cep6x*, *cep5x*, *cepr1/2*, and *cepr1/2/rlk7*, it is not clear whether the observed responses are mediated through the CEP4-RLK7 pathway, the CEP-CEPR pathway, or both. In experiments using only *cep6x* and *cepr1/2/rlk7*, it is not possible to distinguish whether it is the RLK7 pathway or the CEPR pathway that is activated. This is especially important in the experiment in Fig. 5. If CEP4 induction and activation of the RLK7 pathway under low nitrogen is sufficient for upregulation of flg22-induced MPK expression under low nitrogen, then the claim that the CEP family peptides regulate immunity under low nitrogen is misleading to readers. In that case, it should be rephrased as CEP4-RLK7 regulates immunity under low nitrogen. To make the whole story clear, please provide data for the question mark in the table attached, even if it is a negative result.

	Pto COR-growth	Pto DC3000 growth	flg22-induced MPK expression at low N	flg22-induced FRK1 expression at low N
cep6x	Increased	Increased	Decreased	Decreased
cep5x (-CEP4)	Increased	?	?	?
cepr1/2	Increased	Increased	?	?
cepr1/2/rlk7	Increased	More increased	Decreased	?
	Fig. 1I-J Fig. 2A Fig. S4B Fig. S8H	Fig. 3M Fig. S4A	Fig. 5D	Fig. 5F

>>> Thank you very much for the constructive comments on our manuscript. We now provide additional data for reduced N-induced promotion of flg22-triggered MAPK activation, comparing *cepr1/2/rlk7*^{AEQ} with *cepr1/2*^{AEQ} and the Col-0^{AEQ} control, as well as comparing *cep6x* with *cep5x* and the Col-0 control. We observed that the promotional effect of reduced N on flg22-induced MAPK activation is compromised in both lower order mutants, suggesting that CEP4 and RLK7 play a minor role in this context. We included this data in Fig. 5C-F. This is indeed quite interesting, suggesting that the enhancing effect of reduced N on FLS2 signalling is primarily dependent on canonical CEPs and CEPRs, while data presented in the other figures highlights the synergistic effect of multiple CEPs (including CEP4) and CEPR1/CEPR2/RLK7 on immunity under normal growth conditions. We discussed these new findings in the manuscript.

We also attempted to repeat reduced N-promoted flg22-induced *FRK1* expression using different mutant combinations, as requested. However, we observed that the initial phenotype in the Col-0 and *cep6x* mutant (repeated 5 times with similar results for the previous submission, Fig. A, black symbols) was not reproducible and very variable among subsequent repeats (Fig. B, orange symbols; Fig. C, combined data from Fig. A and B). A possible explanation is a change in the growth chamber that was used for the experiments, which occurred about one year ago. However, the promotional effect of reduced N on flg22-induced MAPK activation was highly reproducible and remains an integral part of the manuscript.

In addition, to complement the claim that CEP enhances immunity under low nitrogen, I would like to see the sensitivity of four mutants, *cep6x*, *cep5x*, *cepr1/2*, and *cepr1/2/rlk7*, to Pto COR- or Pto DC3000 under low nitrogen.

>>> We now performed the requested experiment, comparing the respective mutants with their wild-type control using *Pto*^{COR-} infection upon flood inoculation of 1/2 MS-phytagel grown plants. We observed that under these conditions, reduced N content generally promoted resistance in the wild-type controls, while this promotional effect was reduced in both *cepr1/2*^{AEQ} and *cepr1/2/rlk7*^{AEQ} (Fig. 5G). Similar to the newly added reduced N MAPK data, this suggests that low N-mediated promotion of PTI is primarily dependent on canonical CEP-CEPR signalling. Similarly, we tested *cep6x* and *cep5x* mutants under these conditions. We observed slightly enhanced resistance under lower N in Col-0, which was compromised in both *cep6x* and *cep5x* (Supplementary Fig. 12J). Taken together with our MAPK data discussed above, this suggests that the promotional effect of reduced N on PTI is primarily dependent on canonical CEPs.

Below is a table (as indicated in the referee comment above) summarizing the phenotypic data of different *cep* and *cepr* mutant combinations and in which figure of the manuscript they can be found. We filled the relevant question marks and also added the requested bacterial infection data under reduced N growth conditions.

Genotype	Normal growth conditions		Reduced N conditions	
	Pto ^{COR-} growth	Pto DC3000 growth	Promotion of flg22-induced MPK activation	Decrease in Pto ^{COR-} growth
cep5x CEP4 WT	Increased (Sup. Fig. 4B)	Mildly increased (not significant) (Sup. Fig. 4C)	Decreased (Fig. 5C, E)	Compromised (Sup. Fig. 12J)

cep6x	Increased (Sup. Fig. 4B)	Increased (Sup. Fig. 4C)	Decreased (Fig. 5C, E)	Compromised (Sup. Fig. 12J)
cepr1/2^{A_{EQ}}	Increased (Sup. Fig. 8H)	Increased (Fig. 3 M)	Decreased (Fig. 5D, F)	Compromised (Fig. 5G)
cepr1/2/rlk7^{A_{EQ}}	Increased (Sup. Fig. 8H)	Further increased (Fig. 3 M)	Decreased (Fig. 5D, F)	Compromised (Fig. 5G)

Fig. 5C is not explained in the text, but is it the roots or the leaves where the expression level of CEP4 was quantified? If it is the roots, the authors should retest the flg22-dependent MPK expression on the *cep6x* plants as rootstock and as grafted scion under low nitrogen conditions.

>>> Fig. 5C depicted *CEP4* expression levels in whole seedlings, comparing exposure to 10% or 100% N MS medium. However, since novel data obtained during revision suggests that *CEP4* plays a minor (if any) role in the N-induced promotion of PTI, we decided to remove this data from the manuscript.

For Fig. 5D, the expressions 100% N and 10% N are confusing, please use something like XX mM N.

>>> We agree that this may be confusing and highlighted the actual N concentrations in all respective figures.

Comments on the response to Reviewer #1:

The authors have generally answered referee #1's comments accurately. However, in some experiments such as Fig. 5, immune response is only shown for a few selected mutants. To conclude that "CEPs and their receptors promote immunity in an N status-dependent manner", it is necessary to examine the response for all mutants (*cep6x*, *cep5x*, *cepr1/2*, and *cepr1/2/rlk7*) in each experiment (see also my comment on this point). Experiments using only *cep6x* and *cepr1/2/rlk7* cannot distinguish whether it is the RLK7 or CEPR pathway that is activated under given conditions.

>>> See comments to Reviewer #2 and the provided table summarizing our findings.

Reviewer #4 (Remarks to the Author):

Phytocytokines are hypothesized to be endogenous peptides that play a pivotal role in integrating diverse external and internal cues to enhance plant health. Among these, C-terminally encoded peptides (CEPs) constitute a class of endogenous peptides known to influence plant growth and participate in the response to nitrogen starvation. However, they have not yet been characterized as phytocytokines involved in plant immunity. In this manuscript, Rzemieniewski et al identified the CEP4 as a novel phytocytokine, recognized by CEP receptors (CEPR1, CEPR2 and RLK7), thereby coordinating plant immunity with nitrogen status. This study proposes that CEPs and their receptors serve as central regulators in adapting biotic stress responses to plant-available resources. Following a round of peer review, the manuscript has undergone improvements in response to reviewer feedback. Nevertheless, certain critical issues raised by reviewers have not been adequately addressed. Addressing these critical issues is crucial for bolstering the reliability of experimental results and amplifying the innovation within the manuscript. These include the "definition issue

regarding CEPs as phyto cytokine,” “whether there are changes in the tissue localization of CEP4, CEPR1, CEPR2, and RLK7 after flg22 treatment or DC3000 infiltration, and the tissue co-localization issue about CEPR receptors with CEP4,” “the promotion of immune response under N starvation conditions is mediated by RLK7 or CEPR” and “the elucidation of mechanistic processes by which CEPs may participate in the crosstalk between low-N and PTI, which constitutes the major highlight of the manuscript.”

Major:

1. Tissue Expression and Mobility of CEP4:

CEP4 is induced by low nitrogen and expressed in the lateral root emergence zone and hydathode region, while CEPR1 and CEPR2 show expression in vasculature and guard cells, respectively. The authors are urged to provide evidence supporting the mobility of CEP4. Moreover, after treatment with flg22 or infection with DC3000, clarification is needed on whether there are changes in the tissue expression of CEP4. Additionally, is there any co-expression of CEPR1, CEPR2, or RLK7 with CEP4 at the tissue level post these treatments?

>>> Thank you very much for your constructive comments. We now tested whether *CEP4* expression patterns, or that of *RLK7/CEPR1/CEPR2*, are changed upon flg22 treatment in shoot tissue using our *promoter::NLS-3xmVenus* lines. However, this was not the case (new Supplementary Fig. 9). We also provide new data, showing that potential expressional overlap exists between *CEP4/RLK7/CEPR2/CEPR1* at the base of lateral root emergence sites (new Supplementary Fig. 10). These may thus represent a tissue in which CEP4 meets all its potential receptors. Testing CEP4 mobility across tissues is a very interesting line of research, but goes beyond the scope of the current manuscript and is planned to be addressed in a future project.

2. Quantitative Analysis of CEP Family Members:

Why weren't all members of the CEP family subjected to quantitative analysis following flg22 treatment or DC3000 infection, particularly CEP12, which shares the highest homology with CEP4? The authors are strongly encouraged to conduct quantitative analyses on all CEP family members in both roots and shoot post these treatments to establish the correlation between tissue-specific expression patterns and their respective functions.

>>> We now include data showing root vs shoot expression of all *CEPs*, comparing flg22 treatment with untreated conditions (Supplementary Fig. 1C). We reveal that the majority of *CEPs* are stronger expressed in roots, but we could detect weak expression (at comparable levels) for all other *CEP* members in shoot tissue as well. Interestingly, there seems to be a pattern that shoot expression of the majority of *CEPs* (including *CEP4*) is mildly increased upon flg22 treatment, while this is not the case in roots. Also, we provide data for all *CEP* expression levels in shoot upon *Pto* DC3000 infection, which again shows, that their expression level is comparable and that the majority of the *CEP* members are mildly upregulated upon infection (Supplementary Fig. 4D).

3. Morphology Phenotypic Assessment Under Low Nitrogen Conditions:

Considering that CEP signaling orchestrates both plant immunity and nitrogen status, there is a need for an elucidation of the morphology phenotypic outcomes resulting from DC3000 inoculation or flg22 treatment in WT, CEP4-OE, and *cep4* mutant plants under low nitrogen conditions.

>>> See comments to Reviewer 1. Our *cep5x* and *cepr1/2^{AE}Q* mutant data revealed that the enhancing effect of reduced N on flg22-triggered MAPK activation and resistance against *Pto^{COR}* is primarily dependent on canonical CEPs-CEPR1/2 signaling and that CEP4 plays a minor (if any) role for low N-induced promotion of PTI. Therefore, we did not pursue further testing with *cep4* single mutants or *CEP4* overexpression lines.

Minor:

1. The study lacks biochemical evidence supporting the interaction between CEPR1 and CEP4, thereby necessitating additional support for the conclusion that CEPR1 functions as a receptor for CEP4.

>>> We thank the Reviewer for the suggestion. We indeed agree with the Reviewer on that point. Unfortunately, despite our efforts, while we were able to successfully express CEPR1 ECD, we could not purify folded healthy protein to proceed with the binding assays. The aggregated size exclusion chromatography profile of CEPR1 is shown in Supplementary Fig. 6, panel A and B; and stated in the results paragraph lines 175-177 that reads as follows: “Unfortunately, *CEPR1^{ECD}* aggregated, as indicated by the early elution of the bulk sample during SEC analysis (~10min, Supplementary Fig. 6A)”.

To overcome this experimental limitation, we have provided genetic evidence that CEPR1 can be involved in CEP4 perception. The *cepr1-3* mutant is largely insensitive to CEP4-triggered seedling growth inhibition (Supplementary Fig. 5C) and shows reduced *FRK1* expression after CEP4 treatment similarly to *cepr2-4* single mutant (Supplementary Fig. 8A), suggesting their direct connection. Also, it is clear that CEPR1 genetically contributes to disease resistance since *cepr2-4* single mutant did not show a detectable phenotype, while *cepr1/cepr2* mutants are more susceptible (Fig. 2A).

2. The authors may provide an explanation of why RLK7 plays a predominant role in the full CEP4 response from an evolutionary perspective (Lines 217-218).

>>> Thank you for this suggestion. We included this in our discussion.

3. In Lines 45-46, it is advisable to include literature citations introducing the concept of phyto cytokines.

>>> We added the relevant citations as requested.

4. The result description in Lines 85-86 should incorporate corresponding figures.

>>> Thank you for spotting this omission. We included citations of the corresponding figures in the text.

REVIEWERS' COMMENTS

Reviewer #2 (Remarks to the Author):

In response to the comments provided by the reviewers, the authors have included several additional results. The authors provided a comprehensive and accurate response to my comments. I would like to acknowledge the authors for their meticulous response, but the conclusions derived from the new experimental results diverge somewhat from those previously drawn.

The paper, one of the main topics of which is the discovery that CEP4, a CEP family peptide, is recognized by RLK7, a receptor involved in immune responses, initially leads the reviewers to expect that the CEP4-RLK7 pathway is a new pathway linking nitrogen deficiency and immune responses. However, the actual contribution of the newly discovered CEP4-RLK7 pathway is minor, and the final conclusion is that the promotional effect of reduced N on PTI is primarily dependent on canonical CEP-CEPR system. The fact that the contribution of the CEP4-RLK7 pathway is minor suggests that there is an unknown pathway connecting CEP-CEPR signaling and PTI, but the authors do not present any information as to what it is. While I respect the authors for their thorough analysis, I feel that the paper's impact has been weakened by the change in conclusions from the previous manuscript. While most papers published in Nature Communications explain the molecular mechanisms behind the phenomenon, this manuscript merely describes the phenomenon of a weakened immune response when CEP signaling is lost. Nor does the manuscript propose a new concept, since examples are already known in which peptide hormones involved in development, such as GOLVEN-RGI, are also involved in immune responses (EMBO Rep. 23, e53281 (2022)).

Reviewer #4 (Remarks to the Author):

The authors have effectively addressed most of my previous concerns, and I am satisfied with the revised version. No further comments.

REVIEWERS' COMMENTS

Reviewer #2 (Remarks to the Author):

In response to the comments provided by the reviewers, the authors have included several additional results. The authors provided a comprehensive and accurate response to my comments. I would like to acknowledge the authors for their meticulous response, but the conclusions derived from the new experimental results diverge somewhat from those previously drawn.

The paper, one of the main topics of which is the discovery that CEP4, a CEP family peptide, is recognized by RLK7, a receptor involved in immune responses, initially leads the reviewers to expect that the CEP4-RLK7 pathway is a new pathway linking nitrogen deficiency and immune responses.

>>> Thank you for your thoughtful and constructive comments. We appreciate your acknowledgment of our response and the additional results provided. However, we would like to clarify that the primary conclusion of our study was not that CEP4 directly links nitrogen status with immunity. Instead, we report that CEP4 represents a distinct group I CEP ligand. Unlike canonical CEPs, which are perceived exclusively by canonical CEP receptors, CEP4 engages both canonical CEP receptors and RLK7, which we identified as a new CEP4-specific CEP receptor. The key findings of our work include:

- Identification of group I CEPs as immune-regulatory peptides
- Identification of RLK7 as a CEP4-specific CEP receptor, distinguishing CEP4 from other CEP ligands
- Combinatorial role of RLK7 and canonical CEPRs in CEP4 perception
- Distinct receptor and output specificities between canonical group I CEPs and CEP4, a group I CEP with unique sequence features.
- Local function of shoot-expressed CEPs for immune regulation, unlike their root-to-shoot signaling role during N-demand responses.
- Additive effect of canonical CEPs and CEP4 to plant immunity against bacterial infection under normal soil-grown conditions.
- First insights into a molecular cross-talk between N homeostasis and immunity, mediated by CEP-CEPR signaling pathways

We hope this clarification addresses your concerns and highlights the contributions of our findings in the context of plant immunity and nitrogen homeostasis.

However, the actual contribution of the newly discovered CEP4-RLK7 pathway is minor, and the final conclusion is that the promotional effect of reduced N on PTI is primarily dependent on canonical CEP-CEPR system. The fact that the contribution of the CEP4-RLK7 pathway is minor suggests that there is an unknown pathway connecting CEP-CEPR signaling and PTI, but the authors do not present any information as to what it is. While I respect the authors for their thorough analysis, I feel that the paper's impact has been weakened by the change in conclusions from the previous manuscript.

>>> Thank you for your insightful feedback. The minor contribution of the CEP4-RLK7 pathway to reduced N-promoted immunity was indeed a surprising finding during the revision process. However, we believe that this further highlights the complexity of CEP perception and its nuanced role in immune regulation. While our study does not provide details on the precise mechanisms connecting nitrogen signaling and CEP-mediated immune modulation, this limitation would still apply even if the CEP4-RLK7 pathway had demonstrated a larger role in this biological context. The mechanistic basis of CEP-mediated modulation of PTI is a

very interesting question that we intend to address in our future work. In the revised manuscript, we discussed these findings and their implications for the broader CEP-CEPR system and PTI in lines 444–480 of the Discussion section. Also, as outlined above, our manuscript highlights the importance of CEP4-RLK7 for plant immunity under normal (soil-grown) conditions. We hope this clarifies our perspective and strengthens the context of our findings.

While most papers published in Nature Communications explain the molecular mechanisms behind the phenomenon, this manuscript merely describes the phenomenon of a weakened immune response when CEP signaling is lost. Nor does the manuscript propose a new concept, since examples are already known in which peptide hormones involved in development, such as GOLVEN-RGI, are also involved in immune responses (EMBO Rep. 23, e53281 (2022)).

>>> Thank you for your comments. We respectfully disagree with the assessment that our manuscript solely describes the weakened immune response associated with loss of CEP signaling (please see the highlighted new findings above). The manuscript also provides a new concept, which is a CEP-dependent cross-talk between plant immunity and N homeostasis. While the influence of nitrogen nutrition on plant immunity has been documented, the underlying mechanisms have remained elusive. Our study offers the first insights that this interaction is regulated via the CEP-CEPR pathway and its role in cell surface immunity. Though a complete molecular dissection of this cross-talk—whether direct or indirect—was beyond the current scope, our findings establish a new framework for exploring this connection. The significance of these results and the questions they raise for future investigation were outlined in the Discussion section.

Reviewer #4 (Remarks to the Author):

The authors have effectively addressed most of my previous concerns, and I am satisfied with the revised version. No further comments.

>>> Thank you for your thorough review and feedback throughout the revision process. We appreciate your thoughtful insights, which have greatly strengthened the manuscript.